# DAK-UCB: Diversity-Aware Prompt Routing for LLMs and Generative Models

**Donya Jafari**
Sharif University of Technology
`donya.jafari111@sharif.edu`

**Farzan Farnia**
The Chinese University of Hong Kong
`farnia@cse.cuhk.edu.hk`

## Abstract

The expansion of generative AI and LLM services underscores the growing need for adaptive mechanisms to select an appropriate available model to respond to a user's prompts. Recent works have proposed offline and online learning formulations to identify the optimal generative AI model for an input prompt, based solely on maximizing prompt-based fidelity evaluation scores, e.g., CLIP-Score in text-to-image generation. However, such fidelity-based selection methods overlook the diversity of generated outputs, and hence, they can fail to address potential diversity shortcomings in the generated responses. In this paper, we introduce the *Diversity-Aware Kernelized Upper Confidence Bound (DAK-UCB)* method as a contextual bandit algorithm for the online selection of generative models with diversity considerations. The proposed DAK-UCB method incorporates both fidelity and diversity-related metrics into the selection process. We design this framework based on prompt-aware diversity score functions that decompose to a two-sample-based expectation over prompt-output pairs in the previous generation rounds. Specifically, we illustrate the application of our framework using joint kernel distance and kernel entropy measures. Our experimental results demonstrate the effectiveness of DAK-UCB in promoting diversity-aware model selection while maintaining fidelity in the generations for a sequence of prompts. The code is available at `https://github.com/Donya-Jafari/DAK-UCB`.

## 1 Introduction

The past few years have witnessed a rapid surge in generative AI services capable of addressing a wide array of tasks, ranging from large language models (LLMs) answering arbitrary questions to text-to-image and video models generating visual content guided by user prompts. Given the growing number of available generative models, a key challenge is how to effectively select suitable generative models for a sequence of user-provided prompts. A conventional approach is to compute an overall evaluation score for each candidate generative AI model and subsequently select the model with the highest aggregate score to address all future prompts. However, this approach implicitly assumes that a single model consistently outperforms the other models across all possible prompts. This assumption has been demonstrated to be untrue in realistic scenarios where different models may excel on different topics or prompt categories (Hu et al., 2025a; Frick et al., 2025).

To address this limitation, recent literature has introduced prompt-aware model selection mechanisms. These methods include offline learning algorithms (Qin et al., 2024; Chen et al., 2024; Frick et al., 2025), which train a selector model using a batch of pre-collected responses of models to prompts as training data. Also, Hu et al. (2025a) propose the online learning PAK-UCB method, formulating the selection task as a contextual multi-armed bandit problem to utilize the user's observed model performances in the previous rounds.

Despite the development of several model selection approaches, the existing methods focus only on the fidelity scores in data generation, while overlooking the diversity of generated samples. For example, in text-to-image generation tasks, existing frameworks evaluate models based on the alignment of the input prompt and generated image, without considering diversity in the image outputs. Such a diversity-unaware selection can lead to output samples that, although individually aligned well with prompts, collectively lack diversity. In addition, overlooking the diversity of output data

**Candidate Generative Models**

Figure 1: Comparison of baseline Kernelized-UCB model selection (CLIP-Score fidelity metric) (Hu et al., 2025a) vs. our proposed diversity-aware DAK-UCB over $T = 500$ rounds. While the baseline Kernelized-UCB does not favor model $G_2$ with higher diversity over model $G_1$, DAK-UCB selected the more diverse $G_2$ more frequently.

can potentially lead to a restricted representation of sensitive attributes, such as gender or ethnicity, in generated datasets. Figure 1 displays an example, where the diversity-unaware baseline kernelized UCB selection algorithm in (Hu et al., 2025a) chooses the less and more diverse generative models (conditioned to "young male" and none) with similar frequencies, ignoring the diversity factor in the selection process. This limitation arises because PAK-UCB, and standard contextual bandit methods more broadly, compute rewards using the *mean* of sample-level scores. Diversity, however, is a *group-level* property determined by the relative positioning of multiple samples, which cannot be expressed through a simple average of individual rewards.

These limitations highlight the importance of diversity-aware selection methods, which explicitly incorporate considerations of output diversity into the model selection process. In this work, we specifically focus on the online selection task, proposing an algorithm designed to leverage previously generated data to select generative models that achieve an optimal balance between fidelity and diversity. Our proposed method, which we call *Diversity-Aware Kernelized Upper Confidence Bound (DAK-UCB)*, extends the kernelized UCB framework (Valko et al., 2013; Hu et al., 2025a) by integrating a diversity-oriented term, in the form of the expectation of a two-sample (prompt,output) random variable, into the contextual bandit objective.

A key challenge in designing DAK-UCB is to determine which diversity scores are compatible with the contextual bandit selection framework. Specifically, we identify a family of joint kernel scores—including prompt-conditional extensions of kernel distance (Bińkowski et al., 2018), RKE (Jalali et al., 2023), and MMD (Gretton et al., 2012)—that can be expressed as expectations of two-sample quadratic forms over prompts and outputs. This structure is central to DAK-UCB: it enables fast-converging estimation from streaming data via kernel ridge regression, yielding principled con-

fidence bounds in the UCB process. Moreover, by combining joint kernel scores with task-specific fidelity metrics (e.g., CLIPScore in text-to-image generation), DAK-UCB can be extended to provide a unified approach to prompt-adaptive selection that balances the fidelity and diversity factors.

Figure 1 shows an application of DAK-UCB for a diversity-ware generative model selection in response to $T = 500$ prompts of MS-COCO dataset (Lin et al., 2014) on generating human-related scenes. The candidate model $G_1$ represents the Stable Diffusion XL (SD-XL) (Stability-AI, 2023) model conditioned to "young male individual", whereas candidate model $G_2$ outputs the SD-XL outputs without any conditioning. While the baseline kernelized UCB fidelity-based selection did not favor the more diverse model $G_2$, generating samples from both models with equal probabilities, the DAK-UCB model selection with the joint RKE diversity score chose model $G_1$ more often over the 500 online selection iterations.

Beyond deterministic prompt-to-model assignment at every iteration of DAK-UCB, DAK-UCB can be adapted to assign the model to an input prompt based on a non-degenerate mixture of the models. As noted by Rezaei et al. (2025) in the unconditional setting, the optimal diversity-aware selection strategy can itself be a non-degenerate mixture of models. Extending this insight to the conditional, prompt-aware setting, a mixture-based selector effectively rolls a biased $m$-sided die to determine which of the $m$ models is queried for a given prompt. We introduce the *Mixture-DAK-UCB* method to realize this idea: an online algorithm that optimizes prompt-dependent mixture probabilities. Mixture-DAK-UCB generalizes the prompt-free Mixture-UCB framework of Rezaei et al. (2025) to the conditional case, enabling diversity-enhancing mixtures tailored to incoming prompts and yielding further improvements in diversity metrics.

We empirically evaluate different variants of the proposed DAK-UCB and Mixture-DAK-UCB algorithms on text-to-image and language model generation tasks. Our results demonstrate improvements in diversity and overall correctness metrics relative to existing contextual bandit algorithms, such as Kernelized UCB, PAK-UCB, and randomized selection strategies. We also validated the defined Joint-RKE and Joint-KD measures for capturing diversity and distributional matching characteristics of prompt-guided generative models. Here we summarize the work's main contributions:

- Studying the role of diversity in prompt-aware selection of generative AI models,
- Introducing the Diversity-Aware Kernelized UCB (DAK-UCB) algorithm, a contextual bandit approach explicitly accounting for the diversity factor in model selection,
- Extending deterministic DAK-UCB selection to prompt-conditioned mixture selection,
- Demonstrating numerical effectiveness of DAK-UCB on several text-to-image generation tasks.

## 2 RELATED WORKS

**Contextual Bandits.** Contextual bandits (CB) extend the multi-armed bandit (MAB) framework by incorporating the context variable to guide the arm selection process (Langford & Zhang, 2007; Foster et al., 2018). A widely-studied CB is the linear CB, which assumes that the expected reward of each arm is a linear function of context (Li et al., 2010; Chu et al., 2011). Kernelized CBs generalize to non-linear reward models by using kernel methods to capture more complex dependencies between contexts and rewards (Valko et al., 2013). Due to the computational cost of kernel methods, recent works have explored approximations using relevant assumptions on the kernel (Calandriello et al., 2019; 2020; Zenati et al., 2022).

To address exploration in linear CBs more effectively, (Abbasi-Yadkori et al., 2011) propose tighter confidence sets using martingale inequalities, leading to stronger theoretical guarantees and improved empirical performance. Moving beyond linearity, Hu et al. (2025c) introduce PromptWise, a multi-iteration-per-round cost-aware contextual bandit for prompt routing in LLMs and generative models. Also, Kveton et al. (2020) propose two randomized exploration algorithms for generalized linear bandits,which leverage Laplace approximations and perturbations of past data to efficiently explore under non-linear models. However, the above CB methodologies do not target diversity awareness in the online learning setting.

**Diversity/Novelty Evaluation Scores and Guidance in Generative Models.** Several methods have been proposed for evaluating and improving the diversity of generative and diffusion models. On the diversity evaluation, the metrics Recall (Sajjadi et al., 2018; Kynkäänniemi et al., 2019), Coverage

(Naeem et al., 2020), Vendi (Dan Friedman & Dieng, 2023; Ospanov et al., 2024; Ospanov & Farnia, 2025), and RKE (Jalali et al., 2023) have been proposed for unconditional (prompt-free) sample generation, and Conditional Vendi/RKE (Jalali et al., 2026; 2025a) and Scendi (Ospanov et al., 2025) have been suggested for prompt-aware diversity measurement. We note that (Zhang et al., 2024; 2025) propose entropy-based measures for novelty of generative models and their comparison, and (Jalali et al., 2025b; Gong et al., 2025) study kernel-based comparison of embeddings.

For guiding sample generation, (Miao et al., 2024) employed reinforcement learning with a diversity reward function in the generation process. (Sehwag et al., 2022) proposed sampling from low-density regions of the data manifold to encourage diverse outputs. (Corso et al., 2024) introduced a particle-based potential function that explicitly maximizes pairwise dissimilarity. Sadat et al. (2024) explored the addition of Gaussian noise to conditioning inputs during inference to promote variability. Lu et al. (2024) developed ProCreate, a distance-based guidance technique. Askari Hemmat et al. (2024); Jalali et al. (2025a) proposed Vendi/Conditional-RKE Score Guidance, which incorporates diversity score guidance in diffusion models. Similarly, Sani et al. (2026) propose MMD guidance to align the diffusion model to a target distribution by minimizing the MMD distance. We highlight that these works aim to improve the diversity and alignment over the sample generation process, unlike our work on the diversity-aware online selection of pre-trained models.

**Multi-Armed Bandit for diversity-based selection.** In a related work, Rezaei et al. (2025) propose Mixture-UCB, a bandit algorithm for selecting mixtures of generative models to maximize diversity, while their proposed approach is not prompt-aware and therefore not applicable to prompt-guided sample generation. (Hu et al., 2025b), (Chen et al., 2025), (Yang et al., 2024), and (Hou et al., 2024) improve the best arm identification by multi-objective optimization, regret minimization, and sample efficiency. Sani et al. (2012) introduce a framework for risk-averse decision-making in bandit problems by integrating variance-sensitive utility functions into exploration strategies. Weinberger & Yemini (2023) study bandits with self-information-based rewards, proposing algorithms that leverage information-theoretic concepts to balance exploration and exploitation. Zhu & Tan (2020) develop Thompson Sampling algorithms for mean-variance bandits, optimizing both expected returns and reward variability. We note that our work focuses on diversity in a contextual bandit setting, where the prompt plays the role of the context, which is not the case in these works' MAB setting.

## 3 PRELIMINARIES

### 3.1 NOTATIONS AND DEFINITIONS

Throughout the paper, we define a conditional generative model $\mathcal{G}$ as a conditional distribution $P_{\mathcal{G}}(x|t)$ where $x \in \mathcal{X}$ is the generated data variable conditioned to the randomly-observed prompt $t \in \mathcal{T}$. Following this definition, every sample generation of model $\mathcal{G}$ is conditioned on a user's provided prompt $T = t$ and then drawing a sample from the conditioned distribution $P_{\mathcal{G}}(x|T = t)$.

### 3.2 KERNEL-BASED SCORES FOR GENERATIVE MODELS

In a sample space $\mathcal{X}$, we call $k : \mathcal{X} \times \mathcal{X} \to \mathbb{R}$ a kernel function if there exists a feature map $\phi : \mathcal{X} \to \mathcal{H}_k$ such that for every $x, x' \in \mathcal{X}$ we have $k(x, x') = \langle \phi(x), \phi(x') \rangle$ where $\langle \cdot, \cdot \rangle$ denotes the inner product in the Hilbert space $\mathcal{H}_k$ of kernel function $k$. Examples of kernel functions include the degree-$r$ polynomial kernel $k_{\text{poly}(r)}(x, y) = (1 + \gamma \langle x, y \rangle)^r$ with parameter $\gamma > 0$ and the RBF (Gaussian) kernel with parameter $\sigma$ defined as:

$$k_{\text{gaussian}(\sigma)}(x, y) = \exp\left(-\frac{\|x - y\|^2}{2\sigma^2}\right)$$

Given a kernel function $k$, we can define the $n \times n$ kernel matrix $K = [k(x_i, x_j)]_{1 \le i,j \le n}$ for $n$ samples $x_1, \ldots, x_n \in \mathcal{X}$. Note that every valid kernel function will result in a positive semi-definite (PSD) kernel matrix for every set of samples. In our analysis, we use the following kernel-based scores and their variants in the online selection process:

- **Maximum Mean Discrepancy (MMD) and Kernel Distance (KD)**: For two probability distributions $P, Q$ on sample space $\mathcal{X}$, (Bińkowski et al., 2018) consider the kernel distance (KD) between $P$ and $Q$ as the square of the maximum mean discrepancy (MMD) Gretton et al. (2012), i.e.,

$$\text{KD}(P, Q) := \mathbb{E}_{x,x' \stackrel{\text{iid}}{\sim} P}[k(x, x')] + \mathbb{E}_{y,y' \stackrel{\text{iid}}{\sim} Q}[k(y, y')] - 2 \cdot \mathbb{E}_{x,y \stackrel{\text{ind}}{\sim} P \times Q}[k(x, y)] \tag{1}$$

In the above definition, the samples $x, x' \sim P_X, y, y' \sim Q_X$ are drawn independently according to the specified distributions.

- **Rényi Kernel Entropy (RKE)**: For probability model $P$ on space $\mathcal{X}$, the Rényi kernel entropy (RKE) (Jalali et al., 2023) is defined as the order-2 Rényi entropy of the normalized population kernel matrix, which reduces to

$$\text{RKE}(P_X) = 1 \Big/ \mathbb{E}_{x, x' \overset{\text{iid}}{\sim} P}[k(x, x')^2] \tag{2}$$

Considering the empirical samples $x_1, \ldots, x_n \sim P$, the empirical RKE score reduces to $\text{RKE}(x_1, \ldots, x_n) = \|\frac{1}{n} K\|_F^{-2}$.

## 4 DIVERSITY-AWARE KERNELIZED UPPER-CONFIDENCE BOUND

To develop a diversity-aware online selection of conditional generative models, we first propose two-sample-based extensions of the KD and RKE scores to the conditional sample generation case. Subsequently, we extend the standard Kernelized-UCB online learning framework by including an upper confidence bound of the joint proposed score functions.

### 4.1 EXTENSION OF KD AND RKE SCORES TO CONDITIONAL GENERATIVE MODELS

We propose the following extensions of the KD in equation 1 and RKE in equation 2 to the conditional sample generation task. Both the extensions in the following apply the original scores to the joint (prompt $t$, data $x$) variable, by using the *product kernel function* $k_{\text{joint}}([t, x], [t', x']) = k_{\text{text}}(t, t') \cdot k_{\text{data}}(x, x')$. As demonstrated by Bamberger et al. (2022); Wu et al. (2025), the product kernel function corresponds to the Hilbert space of the tensor product of the (embedded) prompt and data vectors, effectively capturing the clusters in the dataset of the joint prompt,data vectors.

**Joint Kernel Distance (JKD) distribution matching score.** We propose the following extension of the marginal (prompt-unaware) kernel distance in equation 1 to the prompt-aware kernel distance, which we call *Joint Kernel Distance (JKD)*, for two conditional distributions $P_{X|T}$ and $Q_{X|T}$:

$$\text{JKD}(P_{X|T}, Q_{X|T}) := \text{KD}(P_T \cdot P_{X|T}, P_T \cdot Q_{X|T}) \tag{3}$$

$$= \mathbb{E}_{t, t' \sim P_T, x, x', y, y' \overset{\text{ind}}{\sim} P_{X|T=t} \cdot P_{X|T=t'} \cdot Q_{X|T=t} \cdot Q_{X|T=t'}} \Big[ k_{\mathcal{T}}(t, t')$$

$$\times \Big( k_{\mathcal{X}}(x, x') + k_{\mathcal{X}}(y, y') - k_{\mathcal{X}}(x, y') - k_{\mathcal{X}}(x', y) \Big) \Big]$$

where $k_{\mathcal{T}} : \mathcal{T} \times \mathcal{T} \to \mathbb{R}$ and $k_{\mathcal{X}} : \mathcal{X} \times \mathcal{X} \to \mathbb{R}$ denote the kernel functions for the input prompt $t$ and output $x$, and $P_T$ is a reference distribution on the input variable $T$ (i.e., prompt) over space $\mathcal{T}$. Importantly, the empirical estimation of the expectation in equation 3 can be performed by accessing only one sample generated by $P_{X|T}$ for each input prompt $t$.

**Joint RKE (JRKE) diversity score.** Similarly, we propose the following definition for the joint (prompt-data) RKE score, which we call Joint-RKE (JRKE) score. JRKE is defined to be the RKE score of the joint sample $(T, X) \sim P_T \cdot P_{X|T}$ given a reference prompt distribution $P_T$:

$$\text{JRKE}(P_{X|T}) := \text{RKE}(P_T \cdot P_{X|T}) \tag{4}$$

$$= 1 \Big/ \mathbb{E}_{t, t' \overset{\text{iid}}{\sim} P_T, x, x' \overset{\text{ind}}{\sim} P_{X|T=t} \cdot P_{X|T=t'}} \Big[ k_{\mathcal{T}}(t, t')^2 k_{\mathcal{X}}(x, x')^2 \Big]$$

This score varies monotonically with its inverse, i.e, Inverse-JRKE score denoted by I-JRKE:

$$\text{I-JRKE}(P_{X|T}) := \mathbb{E}_{t, t' \overset{\text{iid}}{\sim} P_T, x, x' \overset{\text{ind}}{\sim} P_{X|T=t} \times P_{X|T=t'}} \Big[ k_{\mathcal{T}}(t, t')^2 k_{\mathcal{X}}(x, x')^2 \Big] \tag{5}$$

Similar to the JKD score, the expectation in the diversity-based Inverse-JRKE score can be estimated using a single output $X \sim P_{X|T=t}$ for every prompt $T = t$.

### 4.2 DIVERSITY-AWARE ONLINE LEARNING VIA DAK-UCB

To propose a diversity-aware online selection framework, we leverage our proposed conditional diversity scores in Equations 3 and 5, within the contextual bandit framework. The prompt $t$ serves

as the context, and we seek a policy $\Pi : \mathcal{T} \to [G]$ that balances fidelity and diversity objectives. A key feature of the introduced diversity scores is that they both decompose into expectations of prompt-level functions, enabling online estimation with a single sample per prompt. The following proposition highlights this property of the JKD and Inverse-JRKE scores.

**Proposition 1.** *For conditional distributions $P_{X|T}, Q_{X|T}$ and reference distribution $P_T$:*

*(a) The Inverse-JRKE admits the decomposition:*

$$\text{I-JRKE}(P_{X|T}) = \mathbb{E}_{t \sim P_T, x \sim P_{X|T=t}}[\phi_{\text{I-JRKE}}(t, x)], \tag{6}$$

*where $\phi_{\text{I-JRKE}}(t, x) = \mathbb{E}_{t' \sim P_T, x' \sim P_{X|T=t'}}[k_{\mathcal{T}}(t, t')^2 k_{\mathcal{X}}(x, x')^2]$.*

*(b) The JKD for comparing model $g$ against reference $Q$ admits:*

$$\text{JKD}(P_g, Q) = \mathbb{E}_{t \sim P_T, x \sim P_g(\cdot|t)}[\phi_{\text{JKD}}^{(g)}(t, x)], \tag{7}$$

*where $\phi_{\text{JKD}}^{(g)}(t, x) = \mathbb{E}_{t' \sim P_T}[k_{\mathcal{T}}(t, t')(\mathbb{E}_{x' \sim P_g(\cdot|t')}[k_{\mathcal{X}}(x, x')] - \mathbb{E}_{y' \sim Q(\cdot|t')}[k_{\mathcal{X}}(x, y')])]$.*

Proposition 1 highlights a crucial structural property of the proposed diversity scores: both I-JRKE and JKD admit a *two-sample expectation form*, in which the overall metric decomposes into the expectation of a prompt-level function of a single generated sample. This is important in the online setting, because it ensures that each round of interaction with a model provides an *unbiased stochastic label* for the corresponding diversity function, even though the original metric is defined in terms of expectations over pairs of prompts and outputs. Therefore, the two-sample form makes these scores applicable to the kernelized UCB algorithm, as we can run kernel ridge regression (KRR) on the stochastic labels and obtain confidence bounds that are comparable to those for the fidelity score.

Based on this decomposition, we define for each model $g$ and prompt $t$ prompt-level target functions:

$$s_g(t) := \mathbb{E}_{x \sim P_g(\cdot|t)}[\phi_{\text{fidelity}}(t, x)], \quad D_g(t) := \mathbb{E}_{x \sim P_g(\cdot|t)}[\phi_g(t, x; \mathcal{H}_t)]. \tag{8}$$

Here $\phi_{\text{fidelity}}(t, x)$ denotes a fidelity score of a prompt–output pair, instantiated in our experiments as the CLIP-Score between text prompt $t$ and generated image $x$. The function $\phi_g(t, x; \mathcal{H}_t)$ is a per-sample diversity score, whose expectation recovers the desired diversity metric in Proposition 1. The history $\mathcal{H}_t$ is only used to instantiate reference expectations over past outputs.

At each round $t$, DAK-UCB treats the prompt $p_t$ as context in the per-arm kernelized contextual bandit process (Hu et al., 2025a), and compares arms via a per-arm UCB on the combined objective $J_g(t) = s_g(t) + \lambda D_g(t)$, where $s_g(t)$ (e.g. CLIP-Score in our experiments) is the fidelity score and $D_g(t)$ is defined with $\phi_g$ instantiated as either the (negative) I-JRKE score or the (negative) JKD score as in Proposition 1. After observing a single sample $x_i \sim P_{g_i}(\cdot \mid t_i)$, we form unbiased labels $y_i^{(s)} = \phi_{\text{fid}}(t_i, x_i)$ and $y_i^{(D)} = \psi_{g_i}(t_i, x_i; \mathcal{H}_i)$, update per-arm KRR models for $s_g$ and $D_g$, and select the next arm using an optimistic estimate

$$\widehat{J}_g^{\text{UCB}}(t_i) = (\widehat{s}_g(t_i) + \beta^{(s)}\widehat{\sigma}_g^{(s)}(t_i)) + \lambda(\widehat{D}_g(t_i) - \beta^{(D)}\widehat{\sigma}_g^{(D)}(t_i)),$$

i.e., an upper bound for $s_g$ and a *lower* bound for $D_g$ (since $D_g$ is a signed diversity *reward*, equal to the negative of the underlying penalty). Confidence radii $\beta^{(s)}, \beta^{(D)}$ follow the standard KRR-UCB form as detailed in Algorithm 1.

In Appendix B, we establish a regret bound for a phased variant of our algorithm, Sup-DAK-UCB. This result shows that the known regret guarantees of kernelized UCB methods (Chu et al., 2011; Valko et al., 2013; Hu et al., 2025a) can be systematically extended to our diversity-aware objective. A key technical component of this analysis is that the JRKE and JKD metrics admit the two-sample expectation structure, thereby enabling integration with kernelized-UCB confidence bounds. This structural property is specific to JRKE and JKD and allows us to obtain regret guarantees for diversity-aware model selection. The following provides an informal statement of the resulting regret bound, and the proof is deferred to Appendix B.

**Theorem 1** (Informal regret bound for DAK-UCB). *Under Assumptions 1-3 in Appendix B (normalized kernels, sub-Gaussian noise, and RKHS regularity for $s_g$ and $D_g$), the phased variant* SUP-DAK-UCB *algorithm satisfies the following regret bound where the information-gain $\Gamma_T^{(s)}$ and effective-dimension $\Gamma_T^{(D)}$ terms are defined in Appendix B:*

$$\text{Regret}(T) = \widetilde{\mathcal{O}}\Big(\sqrt{G T \Gamma_T^{(s)}} + \lambda \sqrt{G T \Gamma_T^{(D)}}\Big)$$

---

**Algorithm 1:** Diversity-Aware Kernelized UCB (DAK-UCB)

---

**Input:** $G$ generative models, horizon $T$, prompt distribution $\mathcal{P}$, trade-off $\lambda$, diversity score
$\quad\quad \psi \in \{-\text{I-JRKE}, -\text{JKD}\}$
**Output:** $T$ generated outputs

1 Initialize per-arm KRR estimators for fidelity $s_g$ and diversity $D_g$
2 Sample prompt $t_i \sim \mathcal{P}$
3 **for** $g = 1$ **to** $G$ **do**
4 $\quad$ Predict fidelity and diversity with KRR: $(\widehat{s}_g(t_i), \widehat{\sigma}_g^{(s)}(t_i)), (\widehat{D}_g(t_i), \widehat{\sigma}_g^{(D)}(t_i))$;
5 $\quad$ Form UCB score: $\widehat{J}_g^{\text{UCB}}(t_i) \leftarrow (\widehat{s}_g(t_i) + \beta^{(s)}\widehat{\sigma}_g^{(s)}(t_i)) + \lambda(\widehat{D}_g(t_i) + \beta^{(D)}\widehat{\sigma}_g^{(D)}(t_i))$
6 Select model $g_i \leftarrow \arg\max_g \widehat{J}_g^{\text{UCB}}(t_i)$
7 Generate output $x_i \sim P_{g_i}(\cdot \mid t_i)$
8 Form labels $y_i^{(s)} = \phi_{\text{fid}}(t_i, x_i), y_i^{(D)} = \psi_{g_i}(t_i, x_i; \mathcal{H}_i)$
9 Update KRR models of $g_i$ with $(t_i, y_i^{(s)})$ and $(t_i, y_i^{(D)})$
10 Update history $\mathcal{H}_{i+1} \leftarrow \mathcal{H}_i \cup \{(t_i, x_i, g_i)\}$

---

### 4.3 PROMPT-AWARE MIXTURE SELECTION VIA QUADRATIC OPTIMIZATION

While DAK-UCB selects a single model per prompt, maximizing diversity can require prompt-dependent mixtures of the available models, where we denote the model mixture probability values of prompt $t$ with notation $\boldsymbol{\alpha}(t) \in \Delta_G$. Therefore, for $G$ conditional generation models in $\{P_g(\cdot|t)\}_{g=1}^G$, we consider a prompt-aware mixture $\boldsymbol{\alpha}(t) \in \Delta_G$, yielding $P_{\boldsymbol{\alpha}}(\cdot|t) = \sum_{g=1}^G \alpha_g(t)P_g(\cdot|t)$. We focus here on the I-JRKE diversity penalty; the analogous construction for JKD is deferred to Proposition 2 in Appendix A. Using the product kernel, the I-JRKE admits the quadratic form

$$\text{I-JRKE}(P_{\boldsymbol{\alpha}}) = \mathbb{E}_{t \sim P_T}[\boldsymbol{\alpha}(t)^\top M(t)\boldsymbol{\alpha}(t)],$$

where $M(t) \in [0,1]^{G \times G}$ collects cross-kernel expectations across models. To ensure stability across prompts, we restrict mixtures to a kernel-Lipschitz competitor set

$$\mathcal{A}_\epsilon = \left\{ \boldsymbol{\alpha} : \mathcal{T} \to \Delta_G : \quad \forall t, t', \ \big|k_{\mathcal{T}}(t, t')\big| \cdot \big\|\boldsymbol{\alpha}(t) - \boldsymbol{\alpha}(t')\big\|_1 \leq \epsilon \right\},$$

which guarantees that nearby prompts yield similar mixtures and incurs only an $O(\epsilon)$ approximation error. In the Appendix A, we discuss that, under the above mixture feasible set, an approximate solution follows solving the following problem where at each prompt $t$, the decision rule reduces to the concave quadratic maximization

$$\boldsymbol{\alpha}_t^* = \underset{\boldsymbol{\alpha} \in \Delta_G}{\arg\max} \ \big\langle \boldsymbol{\alpha}, \widehat{s}^{\text{UCB}}(t) \big\rangle - \lambda \, \boldsymbol{\alpha}^\top \widehat{M}^{\text{UCB}}(t)\boldsymbol{\alpha}$$

where $\widehat{s}^{\text{UCB}}(t)$ are fidelity UCB estimates and $\widehat{M}^{\text{UCB}}(t)$ is the projection of the kernelized-UCB estimation of $M(t)$ onto the PSD matrices by zeroing its negative eigenvalues. We call the resulting mixture-model selection method *Mixture-DAK-UCB*, as detailed in Algorithm 2 at Appendix A.

### 5 NUMERICAL RESULTS

We numerically evaluated the proposed DAK-UCB and its mixture variant, Mixture-DAK-UCB, in several experiments. In our numerical experiments on text and image data, we used the CLIP encoder (Radford et al., 2021) as the backbone text embedding and DINOv2 (Oquab et al., 2023) as the image embedding as suggested by Stein et al. (2023). We considered the following online model selection baselines in our evaluation of DAK-UCB and Mixture-DAK-UCB:

- **One Arm Oracle:** The one-arm oracle baseline has knowledge of the evaluation scores of *each individual generative model* (aggregated over the validation prompt set). This baseline universally selects the individual model with the best aggregate score to handle all the prompts.

- **Random Selection:** This baseline randomly selects an arm for an input prompt, where each arm is selected uniformly with equal probability, and the selection across prompts are run independently.

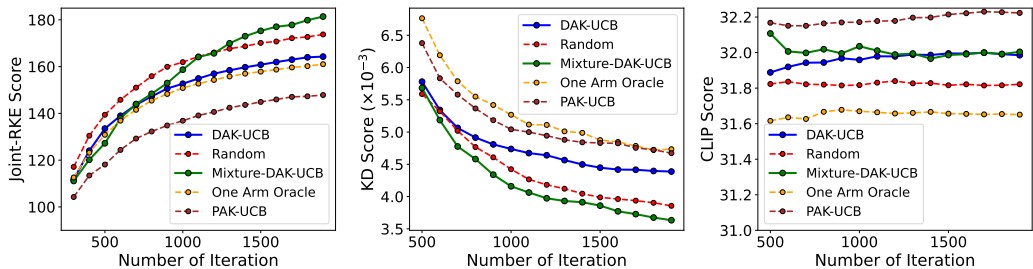

Figure 2: Performance comparison on JKD score and Joint-RKE for MS-COCO prompt clusters using Kandinsky, SDXL, and GigaGAN.

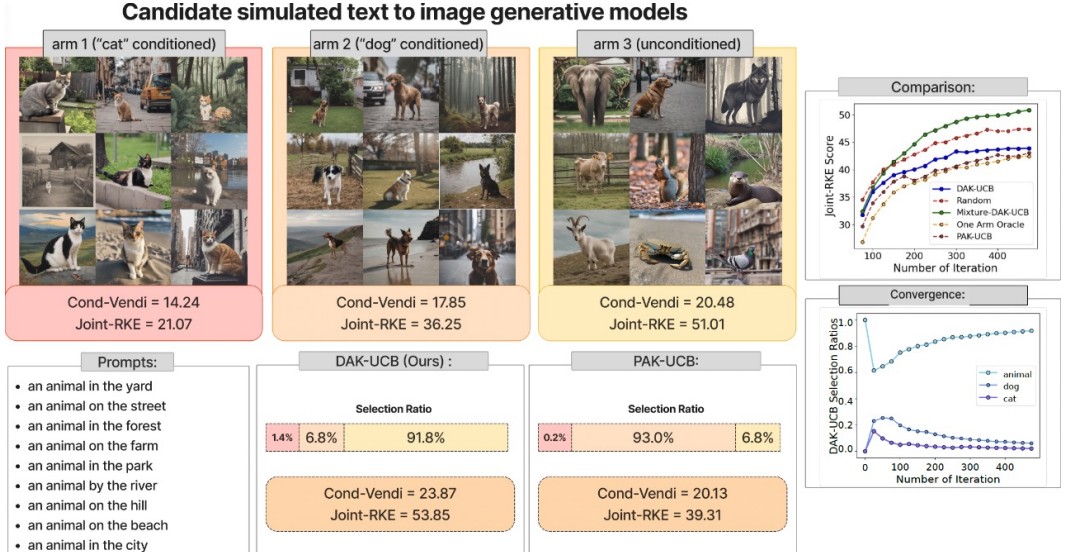

Figure 3: Visualization of simulated generative models with less-diverse Models 1,2 and more-diverse Model 3. DAK-UCB and PAK-UCB selection ratios,scores over 500 rounds are reported.

- **PAK-UCB:** This baseline is a diversity-unaware contextual bandit algorithm (embedded prompt is the context variable), selecting the model only based on the CLIP-score fidelity score in text-to-image generation.

**DAK-UCB applied to diversity-aware text-to-image model selection on MS-COCO prompts.**
We considered the prompts in the MS-COCO (Lin et al., 2014) validation subset. We uniformly sampled a thousand prompts containing the words: cat, dog, car, cake, bowl, bike, tree, airplane, park, and elephant. Three generative models were used as candidate text-to-image generation models in the experiment: Kandinsky (Arkhipkin et al., 2024), SDXL (Stability-AI, 2023), and Giga-GAN (Kang et al., 2023). The experiment ran for 2000 iterations, where at each iteration a random prompt from a random cluster was chosen, and our objective selected the best arm that balanced both diversity and fidelity. The results are averaged over 10 trials to reduce noise from random prompt selection. Figure 2 shows that Mxiture-DAK-UCB could achieve the highest diversity Joint-RKE score. We also used MS-COCO test samples as the reference dataset and report the KD scores with Mixture-DAK-UCB obtaining the best score. Note that the KD metric evaluates both diversity and quality factors.

**Experiment on simulated text-to-image models with varying diversity in "animal" image generation.** In this experiment, we simulated three animal image generation arms, where the first two arms (less-diverse) are outputting the SD-XL generated data conditioned on "cat" and "dog" samples, respectively. On the other hand, the third model (more-diverse) generates the picture of an animal uniformly selected from a list of 10 animals. To run the experiment, we used GPT-4o to

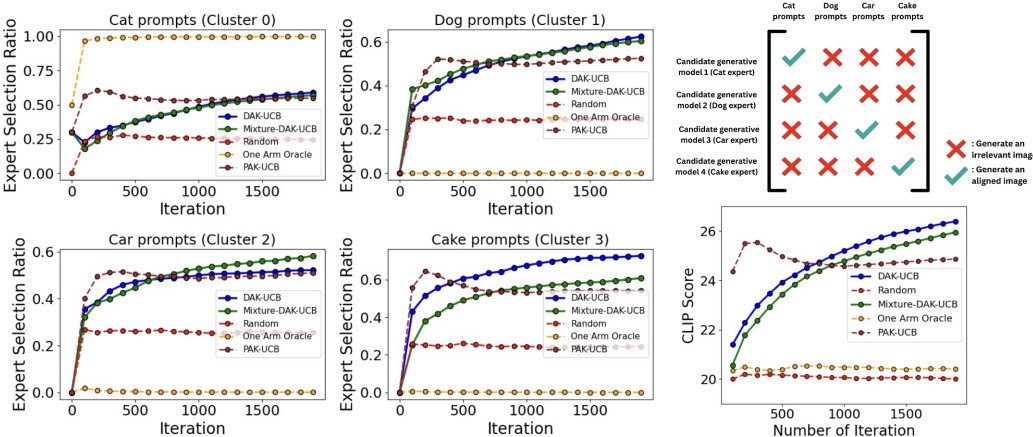

Figure 4: Expert Selection Ratio and Performance Comparison between DAK-UCB and baselines using the JKD score for diversity term in DAK-UCB.

generate 200 independent prompts about "an animal" in different scenes, with sample prompts provided in Figure 3. Figure 3 shows the conditional-Vendi and Joint-RKE scores for each of the three simulated arms, which indicate that the "cat" and "dog" simulated arms were less diverse than the third simulated arm. We ran the DAK-UCB algorithm for 500 iterations, where at each step a random prompt was chosen and the output from the algorithm's selected arm was observed. The results demonstrate that the DAK-UCB tended to generate images from the more diverse third arm, while the CLIP-Score-based PAK-UCB baseline generated samples from the less-diverse second model more frequently.

**Identification of prompt-relevant diversity via DAK-UCB**. In the experiment, we tested DAK-UCB in outputting samples with prompt-relevant diversity. We used these four prompt groups of the experiment of Figure 2 from MS-COCO validation set: "Cat", "Dog", "Car", and "Cake". We designed four arms, each acting as an expert on one of these clusters, where the expert arm on each subject generates aligned samples with the prompt of the same type, while it generates images of a randomly-selected incorrect type for the remaining three subjects For example, ARM1(Cat expert) generates an image using SDXL when given a prompt from the Cat cluster; otherwise, it samples a random irrelevant prompt from other types and generates an image for that prompt using SDXL. In in Figure 4, we report the expert arm selection ratio for each prompt category and the average CLIP-Scores over iterations for each baseline. As suggested by the results in Figure 4, both the JKD-based and Clip-Score+I-JRKE-based DAK-UCB (Appendix, Figure 15) methods could avoid generating prompt-irrelevant output and did not attempt to increase diversity by generating unrelated content.

**Diversity Collapse Across LLMs and the Benefit of Mixtures:** To illustrate the importance of mixture-based selection in realistic language-generation settings, we evaluated three widely used open-source LLMs on a simple iterative generation task: Llama3.2 (AI, 2024), Qwen2 (Team, 2024), and Gemma3 (DeepMind, 2024). At each round, the model produced a short sentence about a vibrant city in North America. Each arm exhibited a persistent and distinct geographic bias: Llama repeatedly focused on New Orleans, Gemma overwhelmingly generated Chicago, and Qwen2 consistently favored New York City. These model-specific collapse modes are visualized in Figure 5.

  Despite their strong capabilities, all three models suffered from diversity collapse, but crucially, their collapse modes were complementary rather than identical. This directly motivates mixture-based selection: although each arm exhibits low diversity on its own, their differing failure modes allow a mixture to achieve significantly higher output diversity. Using the Cond-Vendi metric, we show that the mixture selected by Mixture-DAK-UCB attains substantially higher diversity than any single model. We repeat this experiment for two additional prompts—a vibrant city in Europe" and a renowned celebrity"—and the visualizations in Figure 5 consistently demonstrate the diversity gains unique to model mixtures.

**Additional Numerical Results**. In Appendix C, we also report the numerical results of applying DAK-UCB for diversity-aware model selection in the tasks of prompt-aware selection of simulated LLMs with different diversity scores and the image-to-model assignments of image captioning models. We also present the results of the ablation study for testing the effect of the choice of image embedding and the coefficient of DAK-UCB objective's diversity term.

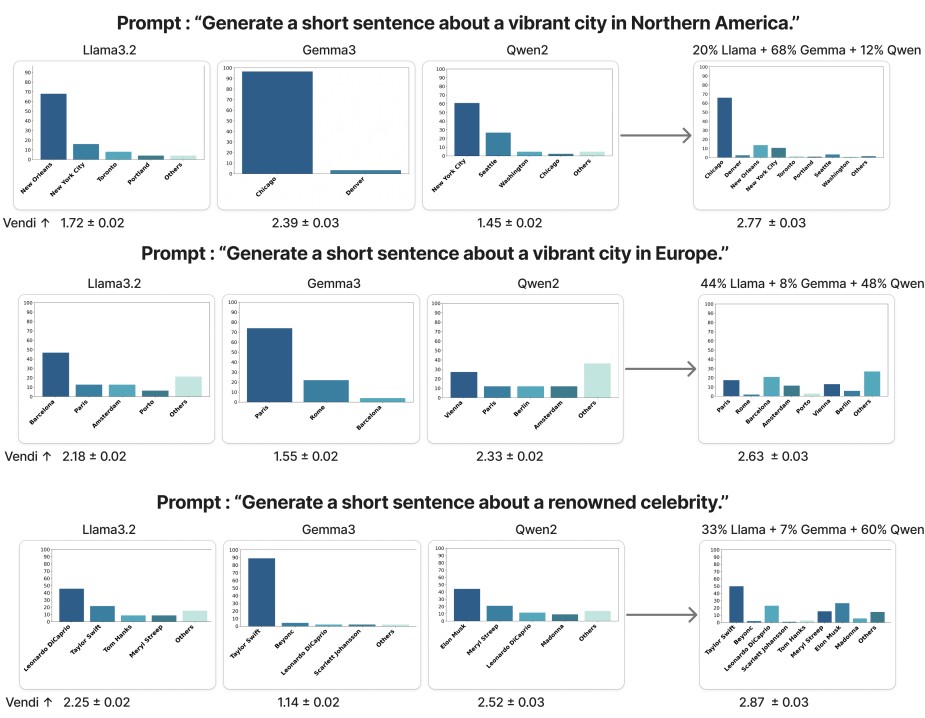

Figure 5: Vendi diversity comparison across Llama3.2, Qwen2, Gemma3, and their mixture for prompts: "Generate a short sentence about a vibrant city in Northern America." ,"Generate a short sentence about a vibrant city in Europe." and "Generate a short sentence about a renowned celebrity."

# 6 CONCLUSION

In this work, we proposed an online learning framework for a diversity-aware prompt-based selection of multiple generative models. Our proposed DAK-UCB can be applied using the defined I-JRKE diversity and JKD correctness scores for improving the diversity factor in sample generation. Notably, the proposed scores reduce to the two-sample expectation over the observed samples, which can be estimated using only one generated sample for an input prompt. In addition, we introduced the Mixture-DAK-UCB extension, which enables optimized prompt-dependent mixtures of generative models and further improves diversity-aware selection.

Beyond text-to-image generation, we also demonstrated applications of DAK-UCB to multi-LLM prompt assignment and image captioning models in Appendix C, illustrating its applicability in broader generative settings. Extending the framework to additional modalities, such as protein, molecular, and graph generative models, is a relevant future direction. The extension of the proposed scores for evaluating and guiding prompt-aware diversity and correctness in data generation will be relevant for future exploration. Also, studying the application of the scores in general bandit settings beyond generative model selection problems is another related future direction.

## ACKNOWLEDGMENTS

The work of Farzan Farnia is partially supported by a grant from the Research Grants Council of the Hong Kong Special Administrative Region, China, Project 14210725, and is partially supported by CUHK Direct Research Grants with CUHK Project No. 4055164 and 4937054. The work is also supported by a grant under 1+1+1 CUHK-CUHK(SZ)-GDSTC Joint Collaboration Fund. Finally, the authors thank the anonymous reviewers and metareviewer for their constructive feedback and suggestions.

## REPRODUCIBILITY STATEMENT

We have taken several steps for the reproducibility of our work. The proposed DAK-UCB and Mixture-DAK-UCB algorithms are fully specified in Section 4 and Appendix A, with pseudocode provided in Algorithms 1–3. Theoretical results are stated with assumptions and proofs in Appendix B. The datasets used in our experiments include standard publicly available benchmarks including MS-COCO as well as synthetically generated prompt sets by the specified GPT-4o model. The details of data selection, prompt construction, model candidates, and evaluation metrics are described in Section 5 and Appendix C. We provide ablation studies in Appendix C.4 to clarify the sensitivity of our results to hyperparameters and embedding choices. An anonymous implementation of our method will be released as supplementary material.

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

## APPENDIX A  DERIVATION OF THE MIXTURE-DAK-UCB PROXY OBJECTIVE FUNCTION

### A.1  UCB FORMULATION OF MIXTURE-DAK-UCB

Building on the mixture objective in the main text, we here prove the approximation guarantee for Mixture-DAK-UCB, and then provide the corresponding UCB formulation.

**Proposition 2.** *Assume the kernel functions are normalized and satisfy $k_{\mathcal{T}}(t,t) = 1$ and $k_{\mathcal{X}}(x,x') = 1$ for all $t,t',x,x'$. For every mixture weight $\boldsymbol{\alpha} \in \mathcal{A}_\epsilon = \left\{ \boldsymbol{\alpha} : \mathcal{T} \to \Delta_G : \forall t,t', \left|k_{\mathcal{T}}(t,t')\right| \cdot \left\|\boldsymbol{\alpha}(t) - \boldsymbol{\alpha}(t')\right\|_1 \leq \epsilon \right\}$, the following hold:*

(a) *For the I-JRKE score of the mixture $P_{\boldsymbol{\alpha}}$ defined as $\text{I-JRKE}(P_{\boldsymbol{\alpha}}) = \mathbb{E}_{t \sim P_T}\left[\boldsymbol{\alpha}(t)^\top M(t)\boldsymbol{\alpha}(t)\right]$, the proxy I-JRKE score $\text{I-JRKE}_{approx}(P_{\boldsymbol{\alpha}}) = \mathbb{E}_t\left[\sum_{g,g'} \alpha_g(t)\alpha_{g'}(t)M_{gg'}^{RKE}(t)\right]$ results in an $\epsilon$-bounded error:*

$$\left|\text{I-JRKE}(P_{\boldsymbol{\alpha}}) - \text{I-JRKE}_{approx}(P_{\boldsymbol{\alpha}})\right| \leq \epsilon$$

(b) *For the JKD score of the mixture $P_{\boldsymbol{\alpha}}$ defined as $\text{JKD}(P_{\boldsymbol{\alpha}}, Q) = \mathbb{E}_{t,t'}\left[k_{\mathcal{T}}(t,t')\sum_{g,g'} \alpha_g(t)\alpha_{g'}(t')K_{gg'}^{JKD}(t,t')\right]$, the proxy JKD score $\text{JKD}_{approx}(P_{\boldsymbol{\alpha}}, Q) = \mathbb{E}_{t,t'}\left[k_{\mathcal{T}}(t,t')\sum_{g,g'} \alpha_g(t)\alpha_{g'}(t)K_{gg'}^{JKD}(t,t')\right]$ results in an $\epsilon$-bounded error:*

$$\left|\text{JKD}(P_{\boldsymbol{\alpha}}, Q) - \text{JKD}_{approx}(P_{\boldsymbol{\alpha}}, Q)\right| \leq \epsilon.$$

*Proof.* **Proof for (a).** Considering the definitions, we have

$$\left| \text{I-JRKE}(P_{\boldsymbol{\alpha}}) - \text{I-JRKE}_{\text{approx}}(P_{\boldsymbol{\alpha}}) \right|$$

$$= \left| \mathbb{E}_{t,t'}\left[ k_{\mathcal{T}}^2(t,t') \sum_{g,g'} \alpha_g(t)\alpha_{g'}(t') K_{gg'}^{\text{RKE}}(t,t') \right] - \mathbb{E}_t\left[ \sum_{g,g'} \alpha_g(t)\alpha_{g'}(t) M_{gg'}^{\text{RKE}}(t) \right] \right|$$

$$= \left| \mathbb{E}_{t,t'}\left[ k_{\mathcal{T}}^2(t,t') \sum_{g,g'} \alpha_g(t)[\alpha_{g'}(t') - \alpha_{g'}(t)] K_{gg'}^{\text{RKE}}(t,t') \right] \right|.$$

For fixed $t, t'$, we bound the inner sum as

$$\left| \sum_{g,g'} \alpha_g(t)[\alpha_{g'}(t') - \alpha_{g'}(t)] K_{gg'}^{\text{RKE}}(t,t') \right| \leq \sum_{g,g'} \alpha_g(t)|\alpha_{g'}(t') - \alpha_{g'}(t)|$$

$$\leq \|\boldsymbol{\alpha}(t') - \boldsymbol{\alpha}(t)\|_1,$$

where the last line uses $\sum_g \alpha_g(t) = 1$. From the Lipschitz condition $\boldsymbol{\alpha} \in \mathcal{A}_\epsilon$:

$$|k_{\mathcal{T}}(t,t')| \cdot \|\boldsymbol{\alpha}(t') - \boldsymbol{\alpha}(t)\|_1 \leq \epsilon.$$

Since $k_{\mathcal{T}}^2(t,t') \leq k_{\mathcal{T}}(t,t)k_{\mathcal{T}}(t',t') = 1$, we have $k_{\mathcal{T}}^2(t,t') \leq |k_{\mathcal{T}}(t,t')|$ and then:

$$k_{\mathcal{T}}^2(t,t') \cdot \|\boldsymbol{\alpha}(t') - \boldsymbol{\alpha}(t)\|_1 \leq |k_{\mathcal{T}}(t,t')| \cdot \|\boldsymbol{\alpha}(t') - \boldsymbol{\alpha}(t)\|_1 \leq \epsilon.$$

Therefore, we can write

$$|\text{I-JRKE}(P_{\boldsymbol{\alpha}}) - \text{I-JRKE}_{\text{approx}}(P_{\boldsymbol{\alpha}})| \leq \mathbb{E}_{t,t'}[k_{\mathcal{T}}^2(t,t') \cdot \|\boldsymbol{\alpha}(t') - \boldsymbol{\alpha}(t)\|_1]$$

$$\leq \mathbb{E}_{t,t'}[\epsilon]$$

$$= \epsilon.$$

**Proof for (b).** Using the definitions,

$$\left| \text{JKD}(P_{\boldsymbol{\alpha}}, Q) - \text{JKD}_{\text{approx}}(P_{\boldsymbol{\alpha}}, Q) \right|$$

$$= \left| \mathbb{E}_{t,t'}\left[ k_{\mathcal{T}}(t,t') \sum_{g,g'} \alpha_g(t)\alpha_{g'}(t') K_{gg'}^{\text{JKD}}(t,t') \right] - \mathbb{E}_t\left[ \sum_{g,g'} \alpha_g(t)\alpha_{g'}(t) M_{gg'}^{\text{JKD}}(t) \right] \right|$$

$$= \left| \mathbb{E}_{t,t'}\left[ k_{\mathcal{T}}(t,t') \sum_{g,g'} \alpha_g(t)[\alpha_{g'}(t') - \alpha_{g'}(t)] K_{gg'}^{\text{JKD}}(t,t') \right] \right|.$$

For fixed $t, t'$, we can bound the inner sum as follows

$$\left| \sum_{g,g'} \alpha_g(t)[\alpha_{g'}(t') - \alpha_{g'}(t)] K_{gg'}^{\text{JKD}}(t,t') \right| \leq \sum_{g,g'} \alpha_g(t)|\alpha_{g'}(t') - \alpha_{g'}(t)|$$

$$\leq \|\boldsymbol{\alpha}(t') - \boldsymbol{\alpha}(t)\|_1.$$

From the Lipschitz condition and noting that $k_{\mathcal{T}}(t,t') \leq |k_{\mathcal{T}}(t,t')|$:

$$k_{\mathcal{T}}(t,t') \cdot \|\boldsymbol{\alpha}(t') - \boldsymbol{\alpha}(t)\|_1 \leq |k_{\mathcal{T}}(t,t')| \cdot \|\boldsymbol{\alpha}(t') - \boldsymbol{\alpha}(t)\|_1 \leq \epsilon.$$

As a result, the following holds

$$|\text{JKD}(P_{\boldsymbol{\alpha}}, Q) - \text{JKD}_{\text{approx}}(P_{\boldsymbol{\alpha}}, Q)| \leq \mathbb{E}_{t,t'}[|k_{\mathcal{T}}(t,t')| \cdot \|\boldsymbol{\alpha}(t') - \boldsymbol{\alpha}(t)\|_1]$$

$$\leq \mathbb{E}_{t,t'}[\epsilon]$$

$$= \epsilon.$$

$\square$

Therefore, to formulte the UCB formulation of Mixture-DAK-UCB, at each round $i$ for prompt $t_i$, the learner maintains UCB predictors for

$$\hat{s}^{\text{UCB}}(t_i) \in \mathbb{R}^G, \qquad \widehat{M}^{\text{UCB}}(t_i) \in \mathbb{R}^{G \times G}, \quad \widehat{M}^{\text{UCB}}(t_i) \succeq 0,$$

---

**Algorithm 2:** Mixture Diversity-Aware Kernelized UCB (MIXTURE-DAK-UCB)

---

**Input:** $G$ models; horizon $T$; prompt distribution $\mathcal{P}$; trade-off $\lambda$; diversity primitive
$\quad\quad\quad \psi \in \{\text{ĦRKE, JKD}\}$; (optional) panel rate $\rho \in [0, 1]$

**Output:** $T$ generated outputs

1  Initialize per-model KRR for fidelity $\{s_g\}_{g=1}^{G}$ and matrix-valued KRR for diversity $\{M_{gg'}\}_{g,g'}$;

2  **for** $i = 1$ **to** $T$ **do**

3  $\quad$ Sample prompt $t_i \sim \mathcal{P}$;

4  $\quad$ For all $g$: $(\widehat{s}_g(t_i), \widehat{\sigma}_g^{(s)}(t_i)) \leftarrow$ KRR-PREDICT$_s(g, t_i)$;

5  $\quad$ For all $(g, g')$: $(\widehat{M}_{gg'}(p_t), \widehat{\sigma}_{gg'}^{(M)}(t_i)) \leftarrow$ KRR-PREDICT$_M(g, g', t_i)$;

6  $\quad$ Set $\widehat{s}_g^{\text{UCB}}(t_i) \leftarrow \widehat{s}_g(t_i) + \beta^{(s)}\widehat{\sigma}_g^{(s)}(t_i)$ for all $g$;

7  $\quad$ Set $\widehat{M}_{gg'}^{\text{LCB}}(t_i) \leftarrow \widehat{M}_{gg'}(t_i) - \beta^{(M)}\widehat{\sigma}_{gg'}^{(M)}(t_i)$; project to PSD if needed;

8  $\quad$ $\boldsymbol{\alpha}_i \leftarrow \arg\max_{\alpha \in \Delta_G} \alpha^\top \widehat{s}^{\text{UCB}}(t_i) - \lambda\, \alpha^\top \widehat{M}^{\text{LCB}}(t_i)\, \alpha$;

9  $\quad$ Sample $g_i \sim \boldsymbol{\alpha}_i$; draw $x_i \sim P_{g_i}(\cdot \mid t_i)$;

10  $\quad$ $y_i^{(s)} \leftarrow \phi_{\text{fid}}(t_i, x_i)$;

11  $\quad$ **if** *panel step with prob.* $\rho$ **then**

12  $\quad\quad$ **for** $g = 1$ **to** $G$ **do**

13  $\quad\quad\quad$ draw $x_i^{(g)} \sim P_g(\cdot \mid t_i)$ (reuse $x_i^{(g_i)} = x_i$)

14  $\quad$ KRR-UPDATE$_s(g_i;\ (t_i, y_i^{(s)}))$;

15  $\quad$ **if** *panel step* **then**

16  $\quad\quad$ Update $\{M_{gg'}\}$ with cross-kernel labels built from $\{x_i^{(g)}\}_{g=1}^{G}$;

17  $\quad$ **else**

18  $\quad\quad$ Update $\{M_{gg'}\}$ using the available pairs;

---

where $\widehat{s}^{\text{UCB}}(t_i)$ collects fidelity UCB scores for each model and $\widehat{M}^{\text{UCB}}(t_i)$ is a PSD estimate of the cross-model diversity matrix $M(t_i)$. The per-prompt mixture decision then follows the concave quadratic program

$$\boldsymbol{\alpha}_i^* = \arg\max_{\boldsymbol{\alpha} \in \Delta_G} \left\{ \langle \boldsymbol{\alpha}, \widehat{s}^{\text{UCB}}(t_i) \rangle - \lambda\, \boldsymbol{\alpha}^\top \widehat{M}^{\text{UCB}}(t_i)\, \boldsymbol{\alpha} \right\}. \tag{9}$$

This UCB objective ensures optimism for fidelity while pessimistically accounting for diversity penalties. The chosen $\boldsymbol{\alpha}_i^*$ specifies a sampling distribution over models, from which the algorithm draws $g_i \sim \text{Multinomial}(\boldsymbol{\alpha}_i^*)$ and obtains $x_i \sim P_{g_i}(\cdot|t_i)$. The resulting procedure, which extends the single-model DAK-UCB to mixture assignments, is summarized in Algorithm 2.

## APPENDIX B  REGRET ANALYSIS OF SUP-DAK-UCB (PHASED VARIANT OF DAK-UCB)

As noted in the literature, the theoretical analysis of the standard kernelized UCB method faces the challenge of potentially statistically correlated model selection at different rounds, which renders standard concentration analysis by means of independent observations inapplicable. To circumvent this challenge, we adopt the standard approach of analyzing a staged variant of the proposed DAK-UCB algorithm, which we call Sup-DAK-UCB. The same technique of analyzing Sup-Kerenlzied-UCB and Sup-PAK-UCB have been applied in the related works (Chu et al., 2011; Valko et al., 2013; Hu et al., 2025a).

In the phased variant of SUP-DAK-UCB, within each arm–stage–target triple, the data used by kernel ridge regression (KRR) are independent. This enables the analytical derivations of confidence bounds and a proper regret decomposition, similar to the analysis in (Valko et al., 2013; Hu et al., 2025a). In the following, we first state the updated setting and assumptions in our theoretical analysis, then present the phased algorithmic structure, followed by the theorems and their proofs.

## B.1 Assumptions in Theoretical Analysis of DAK-UCB

Let $\mathcal{G} = \{1, \ldots, G\}$ be $G$ generative models, $\mathcal{T}$ a prompt space with i.i.d. prompts $t_i \sim P_T$, and $\mathcal{X}$ the output space. At round $t$, the algorithm selects a single model $g_i$. Note that the objective function is the following for a parameter $\lambda \geq 0$: $J_g(t) := s_g(t) - \lambda D_g(t)$

**Assumption 1** (Normalized Prompt and Data Kernel Functions). *The prompt kernel $k_{\mathcal{T}} : \mathcal{T} \times \mathcal{T} \to [-1, 1]$ and the output kernel $k_{\mathcal{X}} : \mathcal{X} \times \mathcal{X} \to [-1, 1]$ are positive definite with $k_{\mathcal{T}}(t, t) = k_{\mathcal{X}}(x, x) = 1$ for all $t \in \mathcal{T}$, $x \in \mathcal{X}$. For $g, g' \in \mathcal{G}$, define*

$$K_{gg'}(p, p') := \mathbb{E}_{x \sim P_g(\cdot|p), \, x' \sim P_{g'}(\cdot|p')}\big[k_{\mathcal{X}}(x, x')^2\big] \in [0, 1].$$

**Assumption 2** (Sub-Gaussian noise in kernel regression). *All scalar observations are conditionally $\sigma$-sub-Gaussian given the history: $\mathbb{E}[\exp(\lambda \varepsilon) \mid \mathcal{H}_i] \leq \exp(\lambda^2 \sigma^2 / 2)$ for all $\lambda \in \mathbb{R}$.*

**Assumption 3** (RKHS boundedness for single-model case). *Let $\mathcal{H}_{\mathcal{T}}$ be the RKHS of $k_{\mathcal{T}}$. Assume $s_g \in \mathcal{H}_{\mathcal{T}}$ with $\|s_g\|_{\mathcal{H}_{\mathcal{T}}} \leq B_s$ and $D_g \in \mathcal{H}_{\mathcal{T}}$ with $\|D_g\|_{\mathcal{H}_{\mathcal{T}}} \leq B_D$ for all $g \in \mathcal{G}$.*

## B.2 Introducing Phased Sup-DAK-UCB Algorithm

As mentioned earlier, we analyze a staged variant of DAK-UCB, called *Sup-DAK-UCB*, possessing $M = \lceil \log_2 T \rceil$ stages. In Sup-DAK-UCB, for each generative model-stage pair $(g, m)$ and target type $\tau \in \{s, D\}$, we maintain a frozen index set $\Psi_g^{m,(\tau)}$. To guarantee independence for diversity labels, we additionally maintain a stage snapshot of the archive of past $(t, x)$ pairs for each arm: at the *beginning* of stage $m$, we freeze $\mathcal{D}_g^m$ and *use this snapshot* to build diversity labels throughout stage $m$. We note that new pairs collected during stage $m$ are not used to form diversity labels in the same stage; They will be only available from stage $m + 1$ onward.

To explain the steps of Sup-DAK-UCB in Algorithm 3, note that at iteration $i$ in stage $m$ with candidate set $\widehat{\mathcal{G}}^m$, we perform the following.

1. For each $g \in \widehat{\mathcal{G}}^m$, we perform KRR predictors $(\widehat{s}_{g,i}^m, \widehat{\sigma}_{g,i}^{m,(s)})$ and $(\widehat{D}_{g,i}^m, \widehat{\sigma}_{g,i}^{m,(D)})$ based on $\Psi_g^{m,(s)}$ and $\Psi_g^{m,(D)}$ respectively.

2. We let $\eta_\sigma := \sigma \sqrt{2 \log(2GMT/\delta)}$ and set

$$\beta^{(s)} := B_s \sqrt{\alpha} + \eta_\sigma, \qquad \beta^{(D)} := B_D \sqrt{\alpha} + \eta_\sigma.$$

   We define the *optimistic* score and the *width* as

$$\widetilde{J}_{g,i}^m := \big(\widehat{s}_{g,i}^m + \beta^{(s)} \widehat{\sigma}_{g,i}^{m,(s)}\big) - \lambda\big(\widehat{D}_{g,i}^m - \beta^{(D)} \widehat{\sigma}_{g,i}^{m,(D)}\big), \qquad w_{g,i}^m := \beta^{(s)} \widehat{\sigma}_{g,i}^{m,(s)} + \lambda \beta^{(D)} \widehat{\sigma}_{g,i}^{m,(D)}.$$

3. The stage selection rule in Sup-DAK-UCB is as follows:

   - If $\max_g w_{g,i}^m \leq T^{-1/2}$, we exploit: $g_i \in \arg\max_{g \in \widehat{\mathcal{G}}^m} \widetilde{J}_{g,i}^m$.
   - Else if $\max_g w_{g,i}^m \leq 2^{1-m}$, we eliminate $\{g : \max_{g'} \widetilde{J}_{g',i}^m - \widetilde{J}_{g,i}^m > 2^{2-m}\}$ and set $m \leftarrow m + 1$.
   - Else (explore), we pick every $g_i$ with $w_{g,i}^m > 2^{1-m}$ and append $t$ to $\Psi_{g_i}^{m,(s)}$ and $\Psi_{g_i}^{m,(D)}$.

4. *Feedback:* For single-model selection, we draw $x_i \sim P_{g_i}(\cdot|t_i)$, observe $y_i^{(s)} = s_{g_i}(t_i) + \varepsilon_i^{(s)}$, and build a *stage-frozen*, unbiased, bounded diversity statistic using $\mathcal{D}_{g_i}^m$:

$$\widehat{d}_i = \frac{1}{\max\{1, |\mathcal{D}_{g_i}^m|\}} \sum_{(t', x') \in \mathcal{D}_{g_i}^m} k_{\mathcal{T}}(t_i, t')^2 \, k_{\mathcal{X}}(x_i, x')^2, \qquad \mathbb{E}\big[\widehat{d}_i \mid t_i, \mathcal{D}_{g_i}^m\big] = D_{g_i}(t_i).$$

   We define the zero-mean diversity noise $\varepsilon_i^{(D)} := \widehat{d}_i - D_{g_i}(t_i)$. Finally, we update the archive $\mathcal{D}_{g_i} \leftarrow \mathcal{D}_{g_i} \cup \{(t_i, x_i)\}$, which will only be snapped at the next stage.

---

**Algorithm 3: Sup-DAK-UCB**

---

**Input:** $G$ models, $T$ rounds, prompt dist. $\mathcal{P}$, trade-off $\lambda$, kernels $k_{\mathcal{T}}, k_{\mathcal{X}}$, ridge $\alpha$, confidence $\delta$
**Output:** $T$ generated outputs

1  Set number of stages $M \leftarrow \lceil \log_2 T \rceil$;
2  Initialize per-stage sets $\Psi_g^{m,(s)}, \Psi_g^{m,(D)} \leftarrow \emptyset$ and archives $\mathcal{D}_g \leftarrow \emptyset$;
3  **for** $i = 1$ **to** $T$ **do**
4      Sample prompt $t_i \sim \mathcal{P}$; set $m \leftarrow 1$, $\widehat{\mathcal{G}}^1 \leftarrow [G]$;
5      Freeze stage snapshots $\mathcal{D}_g^m \leftarrow \mathcal{D}_g$ for all $g$;
6      **repeat**
7          **for** $g \in \widehat{\mathcal{G}}^m$ **do**
8              Compute fidelity and diversity predictions by KRR: $(\hat{s}_g^m, \sigma_g^{m,(s)}), (\hat{D}_g^m, \sigma_g^{m,(D)})$;
9              Form optimistic score $\widetilde{J}_g^m$ and width $w_g^m$;
10         **if** $\max_g w_g^m \leq T^{-1/2}$ **then**
11             $g_i \leftarrow \arg\max_g \widetilde{J}_g^m$;
12             **break**;
13         **else**
14             **if** $\max_g w_g^m \leq 2^{1-m}$ **then**
15                 Eliminate: $\widehat{\mathcal{G}}^{m+1} \leftarrow \{g : \widetilde{J}_g^m \geq \max_h \widetilde{J}_h^m - 2^{2-m}\}$;
16                 $m \leftarrow m + 1$; freeze new snapshots;
17             **else**
18                 Explore: choose $g_i$ with $w_{g_i}^m > 2^{1-m}$;
19                 Append $i$ to $\Psi_{g_i}^{m,(s)}, \Psi_{g_i}^{m,(D)}$;
20                 **break**;
21     **until**;
22     Generate $x_i \sim P_{g_i}(\cdot|t_i)$, observe $y_i^{(s)}$ and stage-frozen $y_i^{(D)}$;
23     Update live archive $\mathcal{D}_{g_i} \leftarrow \mathcal{D}_{g_i} \cup \{(t_i, x_i)\}$;

---

### B.3 KRR NOTATION AND INFORMATION MEASURES

For an index set $\Psi$, let $\Phi_\Psi = [\phi(t_i)^\top]_{i\in\Psi}$, $K_\Psi = \Phi_\Psi \Phi_\Psi^\top$, $k_\Psi(t) = [k_{\mathcal{T}}(t, t_i)]_{i\in\Psi}$, and $A_\Psi := \Phi_\Psi^\top \Phi_\Psi + \alpha I$. The KRR predictor and posterior deviation at $t$ are

$$\widehat{\mu}(t; \Psi) = k_\Psi(t)^\top (K_\Psi + \alpha I)^{-1} y_\Psi = \phi(t)^\top A_\Psi^{-1} \Phi_\Psi^\top y_\Psi, \qquad \widehat{\sigma}^2(t; \Psi) = \phi(t)^\top A_\Psi^{-1} \phi(t).$$

We use the shorthand $\eta_\sigma(\delta) := \sigma\sqrt{2\log(2/\delta)}$. Also, we use the following complexity measures:

$$\gamma(\Psi) := \tfrac{1}{2} \log\det(I + \alpha^{-1} K_\Psi), \qquad \Gamma_T := \max_{\Psi : |\Psi| \leq T} \gamma(\Psi).$$

### B.4 SINGLE-MODEL SELECTION SUP-DAK-UCB REGRET BOUNDS

**Lemma 1.** *Consider arm $g$, stage $m$, and target $\tau \in \{s, D\}$. Consider the sequence of time indices $\{i\}$ that get appended to $\Psi_g^{m,(\tau)}$ by the stage rule. Conditional on the* prompt sequence $\{t_i\}$ *and the stage-frozen archive snapshot $\mathcal{D}_g^m$ (for $\tau = D$), the random variables $\{y_i^{(\tau)}\}_{i\in\Psi_g^{m,(\tau)}}$ are mutually independent and satisfy $\mathbb{E}[y_i^{(\tau)} \mid t_i, \mathcal{D}_g^m] = f_g^{(\tau)}(t_i)$, where $f^{(s)} = s$ and $f^{(D)} = D$. Moreover, $\varepsilon_i^{(D)} = \widehat{d}_i - D_g(t_i)$ is conditionally $1/2$-sub-Gaussian.*

*Proof.* For $\tau = s$, $y_i^{(s)} = s_g(p_i) + \varepsilon_i^{(s)}$ with $\varepsilon_i^{(s)}$ conditionally independent across $i$ given $(\mathcal{F}_{i-1}, t_i)$ by Assumption 2. Thus, $\{y_i^{(s)}\}_{i\in\Psi_g^{m,(s)}}$ are mutually independent given $\{t_i\}$.

For $\tau = D$, by construction we use the *stage-frozen* archive snapshot $\mathcal{D}_g^m$. Given $t_i$ and $\mathcal{D}_g^m$, we draw a fresh $x_i \sim P_g(\cdot \mid t_i)$ independent of $(x_j)_{j<i}$. Each summand inside $\widehat{d_i}$ equals

$$Z_{i,(t',x')} := k_{\mathcal{T}}(t_i, t')^2 \, k_{\mathcal{X}}(x_i, x')^2 \in [0, 1],$$

and these are i.i.d. across $(t', x') \in \mathcal{D}_g^m$ *conditional* on $(t_i, x_i)$. Hence,

$$\mathbb{E}\big[\widehat{d_i} \mid t_i, \mathcal{D}_g^m\big] = \frac{1}{\max\{1, |\mathcal{D}_g^m|\}} \sum_{(t',x') \in \mathcal{D}_g^m} k_{\mathcal{T}}(t_i, t')^2 \, \mathbb{E}_{x_i \sim P_g(\cdot|t_i)}\big[k_{\mathcal{X}}(x_i, x')^2\big] = D_g(t_i).$$

Define $\varepsilon_i^{(D)} := \widehat{d_i} - D_g(p_i)$. Since $\widehat{d_i} \in [0, 1]$ and $D_g(t_i) \in [0, 1]$, Hoeffding's lemma implies $\varepsilon_i^{(D)}$ is conditionally $1/2$-sub-Gaussian. The mutual independence of $\{y_i^{(D)}\}$ over $i \in \Psi_g^{m,(D)}$ holds because each $y_i^{(D)}$ is a function of $(t_i, x_i)$ and the fixed snapshot $\mathcal{D}_g^m$, and $(x_i)$ are independent across $i$. $\qquad\square$

**Lemma 2.** *Fix $(g, m, \tau)$ and a nonempty index set $\Psi := \Psi_g^{m,(\tau)}$ that satisfies Lemma 1. Let $f := f_g^{(\tau)} \in \mathcal{H}_{\mathcal{T}}$ with $\|f\|_{\mathcal{H}_{\mathcal{T}}} \leq B_\tau$, and write $y_\Psi = f(\Phi_\Psi) + \varepsilon_\Psi$, where $\varepsilon_\Psi$ are zero-mean, conditionally independent and $\sigma$-sub-Gaussian (with $\sigma = \sigma$ for fidelity and $\sigma = 1/2$ for diversity). Then, for any $t \in \mathcal{T}$ and any $\delta' \in (0, 1)$, with probability at least $1 - \delta'$*

$$\big|\widehat{\mu}(t; \Psi) - f(t)\big| \leq \underbrace{\eta_\sigma(\delta') \, \widehat{\sigma}(t; \Psi)}_{\text{noise term}} + \underbrace{\sqrt{\alpha} \, B_\tau \, \widehat{\sigma}(t; \Psi)}_{\text{shrinkage bias}},$$

*where $\eta_\sigma(\delta') = \sigma\sqrt{2\log(2/\delta')}$. Equivalently, with $\beta^{(\tau)} := B_\tau \sqrt{\alpha} + \eta_\sigma(\delta')$,*

$$\widehat{s} + \beta^{(s)}\widehat{\sigma}^{(s)} \geq s, \qquad \widehat{D} - \beta^{(D)}\widehat{\sigma}^{(D)} \leq D.$$

*Proof.* Let $A = \Phi_\Psi^\top \Phi_\Psi + \alpha I$. Using the identity $\Phi_\Psi^\top (K_\Psi + \alpha I)^{-1} = (\Phi_\Psi^\top \Phi_\Psi + \alpha I)^{-1}\Phi_\Psi^\top = A^{-1}\Phi_\Psi^\top$, we can write

$$\widehat{\mu}(t; \Psi) - f(t) = \phi(t)^\top A^{-1}\Phi_\Psi^\top \big(f(\Phi_\Psi) + \varepsilon_\Psi\big) - \phi(t)^\top w_f$$

where $w_f \in \mathcal{H}_{\mathcal{T}}$ is the (unique) RKHS representer of $f$, i.e. $f(\cdot) = \langle w_f, \phi(\cdot) \rangle$, with $\|w_f\|_{\mathcal{H}_{\mathcal{T}}} = \|f\|_{\mathcal{H}_{\mathcal{T}}} \leq B_\tau$. Inserting and subtracting $\phi(t)^\top A^{-1}\Phi_\Psi^\top \Phi_\Psi w_f$ gives

$$\widehat{\mu}(t; \Psi) - f(t) = \underbrace{\phi(t)^\top A^{-1}\Phi_\Psi^\top \varepsilon_\Psi}_{\text{Term (I)}} + \underbrace{\phi(t)^\top \big(A^{-1}\Phi_\Psi^\top \Phi_\Psi - I\big) w_f}_{\text{Term (II)}}.$$

**Bounding Term (I):** Let $a := \Phi_\Psi A^{-1}\phi(t) \in \mathbb{R}^{|\Psi|}$. By independence and sub-Gaussianity of $\varepsilon_\Psi$ and the standard bound for linear forms of sub-Gaussian vectors, for any $\delta' \in (0, 1)$,

$$\mathbb{P}\Big(\big|a^\top \varepsilon_\Psi\big| \leq \eta_\sigma(\delta') \, \|a\|_2\Big) \geq 1 - \delta', \quad \eta_\sigma(\delta') = \sigma\sqrt{2\log(2/\delta')}.$$

We bound $\|a\|_2$ by noting that

$$\|a\|_2^2 = \phi(t)^\top A^{-1}\Phi_\Psi^\top \Phi_\Psi A^{-1}\phi(t) \leq \phi(t)^\top A^{-1}\phi(t) = \widehat{\sigma}^2(t; \Psi),$$

since $\Phi_\Psi^\top \Phi_\Psi \preceq A$ implies $A^{-1}\Phi_\Psi^\top \Phi_\Psi A^{-1} \preceq A^{-1}$. Hence, with probability at least $1 - \delta'$,

$$\big|\phi(t)^\top A^{-1}\Phi_\Psi^\top \varepsilon_\Psi\big| \leq \eta_\sigma(\delta') \, \widehat{\sigma}(t; \Psi).$$

**Bounding Term (II):** Since $A = \Phi_\Psi^\top \Phi_\Psi + \alpha I$, we have $A^{-1}\Phi_\Psi^\top \Phi_\Psi - I = -\alpha A^{-1}$. (II) becomes equal to $-\alpha \, \phi(t)^\top A^{-1} w_f$. By Cauchy–Schwarz, we can write

$$\big|\phi(t)^\top A^{-1} w_f\big| \leq \big\|A^{-1/2}\phi(t)\big\|_2 \cdot \big\|A^{-1/2} w_f\big\|_2 \leq \underbrace{\sqrt{\phi(t)^\top A^{-1}\phi(t)}}_{= \, \widehat{\sigma}(t;\Psi)} \cdot \underbrace{\|A^{-1/2}\|}_{\leq \, \alpha^{-1/2}} \cdot \|w_f\|.$$

Thus $\big|\phi(t)^\top \big(A^{-1}\Phi_\Psi^\top \Phi_\Psi - I\big) w_f\big| \leq \sqrt{\alpha} \, \|w_f\| \, \widehat{\sigma}(t; \Psi) \leq \sqrt{\alpha} \, B_\tau \, \widehat{\sigma}(t; \Psi)$. Combining (I) and (II) completes the proof. $\qquad\square$

**Lemma 3.** *Fix any round $i$ and stage $m$. Suppose the confidence radii in Lemma 2 hold for all arms in $\widehat{\mathcal{G}}^m$ (with the choice $\delta' = \delta/(GMT)$ and a union bound across $i, g, m$). Then the following hold simultaneously with probability at least $1 - \delta$:*

1. *$|\widehat{s}_{g,i}^m - s_g(t_i)| \leq \beta^{(s)} \widehat{\sigma}_{g,i}^{(s)}$ and $|\widehat{D}_{g,i}^m - D_g(t_i)| \leq \beta^{(D)} \widehat{\sigma}_{g,i}^{(D)}$ for each $g \in \widehat{\mathcal{G}}^m$.*

2. *The true per-prompt maximizer $g_i^* \in \widehat{\mathcal{G}}^m$ after any elimination step.*

3. *For every $g \in \widehat{\mathcal{G}}^m$, $J_{g_i^*}(t_i) - J_g(t_i) \leq 2^{3-m}$.*

*Proof.* Item 1 follows directly from the application of Lemma 2 to $s$ and $D$ with a union bound over $i, g, m$.

Regarding Items 2 and 3, consider a step in which the algorithm chooses either to exploit or eliminate. Throughout this step, by definition of $w_{g,i}^m$ and the trigger $\max_{g \in \widehat{\mathcal{G}}^m} w_{g,i}^m \leq 2^{1-m}$, we have for every $g \in \widehat{\mathcal{G}}^m$,

$$\beta^{(s)} \widehat{\sigma}_{g,i}^{m,(s)} \leq 2^{1-m}, \qquad \lambda \beta^{(D)} \widehat{\sigma}_{g,i}^{m,(D)} \leq 2^{1-m}. \tag{10}$$

Using Lemma 2, we have the following to hold for every component:

$$s_g(t_i) \leq \widehat{s}_{g,i}^m + \beta^{(s)} \widehat{\sigma}_{g,i}^{m,(s)}, \qquad D_g(t_i) \geq \widehat{D}_{g,i}^m - \beta^{(D)} \widehat{\sigma}_{g,i}^{m,(D)}.$$

Therefore,

$$J_g(t_i) = s_g(t_i) - \lambda D_g(t_i) \leq \underbrace{\widehat{s}_{g,i}^m + \beta^{(s)} \widehat{\sigma}_{g,i}^{m,(s)}}_{\leq \widehat{s}_{g,i}^m + 2^{1-m}} - \lambda \underbrace{\left(\widehat{D}_{g,i}^m - \beta^{(D)} \widehat{\sigma}_{g,i}^{m,(D)}\right)}_{\geq \widehat{D}_{g,i}^m - 2^{1-m}} = \widetilde{J}_{g,i}^m + 2^{2-m}.$$

Similarly,

$$J_g(t_i) \geq \widetilde{J}_{g,i}^m - 2^{2-m}.$$

In particular, for the optimal arm $g_i^*$ and any $g \in \widehat{\mathcal{G}}^m$,

$$\widetilde{J}_{g_i^*,i}^m - \widetilde{J}_{g,i}^m \geq J_{g_i^*}(t_i) - J_g(t_i) - 2^{2-m} - 2^{2-m} \geq -2^{3-m}.$$

Hence $\widetilde{J}_{g_i^*,i}^m$ is at most $2^{3-m}$ below any $\widetilde{J}_{g,i}^m$. During elimination, the algorithm removes only arms whose $\widetilde{J}$ is more than $2^{2-m}$ below the current maximum. Since $2^{3-m} > 2^{2-m}$ is the *two-sided* tolerance and $g_i^*$ attains the *one-sided* tolerance bound, $g_i^*$ cannot be eliminated: this proves (2). Finally, any survivor $g$ satisfies

$$J_{g_i^*}(t_i) - J_g(t_i) \leq \left(\widetilde{J}_{g_i^*,i}^m + 2^{2-m}\right) - \left(\widetilde{J}_{g,i}^m - 2^{2-m}\right) \leq 2^{2-m} + 2^{2-m} = 2^{3-m},$$

which proves (3). $\qquad\square$

**Lemma 4.** *Fix $(g, m, \tau)$ and let $(t_j)_{j \in \Psi}$ be the prompts indexed by $\Psi = \Psi_g^{m,(\tau)}$ in the sorted order. Define $A_0 := \alpha I$ and $A_j := A_{j-1} + \phi(t_j)\phi(t_j)^\top$. Then, the following holds*

$$\sum_{j \in \Psi} \widehat{\sigma}^2(t_j; \Psi_{<i}) = \sum_{j \in \Psi} \phi(t_j)^\top A_{j-1}^{-1} \phi(t_j) \leq 2\gamma(\Psi),$$

*Also, the application of the Cauchy-Schwarz inequality implies that*

$$\sum_{j \in \Psi} \widehat{\sigma}(t_j; \Psi_{<i}) \leq \sqrt{|\Psi| \sum_{j \in \Psi} \widehat{\sigma}^2(t_j; \Psi_{<i})} \leq \sqrt{2 |\Psi| \gamma(\Psi)}.$$

*Proof.* For the determinant of matrices, we can write the following:

$$\det(A_j) = \det(A_{j-1}) \cdot \left(1 + \phi(t_j)^\top A_{j-1}^{-1} \phi(t_j)\right).$$

Therefore,

$$\log \frac{\det(A_j)}{\det(A_{j-1})} = \log\left(1 + \widehat{\sigma}^2(t_j; \Psi_{<i})\right).$$

Summing the above from $j = 1$ to $j = |\Psi|$ shows that

$$\sum_{j \in \Psi} \log\left(1 + \widehat{\sigma}^2(t_j; \Psi_{<i})\right) = \log \frac{\det(A_{|\Psi|})}{\det(A_0)} = \log \det\left(I + \alpha^{-1}\Phi_\Psi \Phi_\Psi^\top\right) = 2\,\gamma(\Psi).$$

Since $0 \leq \widehat{\sigma}^2(t_j; \Psi_{<i}) \leq \alpha^{-1}$ (because $A_{i-1} \succeq \alpha I$), we have $\widehat{\sigma}^2 \leq 1$ when $\alpha \geq 1$. Also, note that for every $x \in [0,1]$, $x \leq 2\log(1+x)$ holds, which implies that $\sum_j \widehat{\sigma}^2(t_j; \Psi_{<i}) \leq 2\sum_j \log(1 + \widehat{\sigma}^2(t_j; \Psi_{<i})) = 2\gamma(\Psi)$. Finally, the linear-sum bound follows directly from the application of the Cauchy-Schwarz inequality. $\qquad\square$

**Theorem 2** (DAK-UCB regret). *Define instance-wise regret to be $r_i = J_{g_i^*}(t_i) - J_{g_i}(t_i)$. Under Assumptions 1–3 and with $\beta^{(s)} = B_s\sqrt{\alpha} + \eta_\sigma$, $\beta^{(D)} = B_D\sqrt{\alpha} + \eta_\sigma$ where $\eta_\sigma = \sigma\sqrt{2\log(2GMT/\delta)}$, the phased DAK-UCB satisfies, with probability at least $1 - \delta$,*

$$\mathrm{Regret}(T) \;\leq\; \widetilde{O}\!\left(\sqrt{GT\,\Gamma_T^{(s)}}\right) \;+\; \lambda\,\widetilde{O}\!\left(\sqrt{GT\,\Gamma_T^{(D)}}\right),$$

*where $\Gamma_T^{(\tau)} := \max_{g,m} \gamma\big(\Psi_g^{m,(\tau)}\big)$. Equivalently, replacing $\Gamma_T^{(\cdot)}$ by an effective-dimension proxy $d_{\mathrm{eff}}^{(\cdot)}$, the bound is $\widetilde{O}\big((1 + \sqrt{\alpha})\sqrt{d_{\mathrm{eff}}^{(s)}GT}\big) + \lambda\,\widetilde{O}\big((1 + \sqrt{\alpha})\sqrt{d_{\mathrm{eff}}^{(D)}GT}\big)$.*

*Proof.* We partition rounds into $\mathcal{T}_0$ (*exploitation*: $\max_g w_{g,i}^{m_i} \leq T^{-1/2}$) and $\mathcal{T}_1$ (*exploration*: a width value greater than $2^{1-m}$). On $\mathcal{T}_0$, by Lemma 3 Part 2 and the exploitation trigger, $r_i \leq 2w_{g_i,i}^{m_i} \leq 2T^{-1/2}$, and therefore $\sum_{i \in \mathcal{T}_0} r_i \leq 2\sqrt{T}$. Subsequently over $\mathcal{T}_1$, we group considering $(g,m)$ as

$$\sum_{i \in \mathcal{T}_1} r_i \;\leq\; \sum_{g=1}^{G}\sum_{m=1}^{M}\sum_{i \in \Psi_g^{m,(s)}} 2^{3-m} = \sum_{g,m} 2^{3-m}\,\big|\Psi_g^{m,(s)}\big|.$$

For each such $i$, the append rule implies $w_{g,i}^m > 2^{1-m}$, i.e., $\beta^{(s)}\widehat{\sigma}_{g,i}^{(s)} + \lambda\beta^{(D)}\widehat{\sigma}_{g,i}^{(D)} > 2^{1-m}$. Summing over $t \in \Psi_g^{m,(s)}$ and applying Lemma 4 and Cauchy–Schwarz to the two targets separately yields

$$
\begin{aligned}
2^{1-m}\big|\Psi_g^{m,(s)}\big| \;&\leq\; \beta^{(s)}\sum_{i \in \Psi_g^{m,(s)}} \widehat{\sigma}_{g,t}^{(s)} \;+\; \lambda\beta^{(D)}\sum_{i \in \Psi_g^{m,(D)}} \widehat{\sigma}_{g,i}^{(D)} \\
&\leq\; \beta^{(s)}\sqrt{2\,\big|\Psi_g^{m,(s)}\big|\,\gamma_g^{m,(s)}} \;+\; \lambda\beta^{(D)}\sqrt{2\,\big|\Psi_g^{m,(D)}\big|\,\gamma_g^{m,(D)}}.
\end{aligned}
$$

In the above, $\gamma_g^{m,(\tau)} := \gamma(\Psi_g^{m,(\tau)})$. Taking the summation over $(g,m)$ pairs and noting $M = \lceil \log_2 T \rceil$ completes the proof of the regret bound. $\qquad\square$

## APPENDIX C ADDITIONAL NUMERICAL RESULTS

### C.1 DAK-UCB APPLIED TO LLM SELECTION PROBLEM

**DAK-UCB for Diversity-Aware LLM Selection Using Synthetic Prompts:** In this experiment, we asked GPT-4o to provide five words as categories: *temple*, *painting*, *market*, *horse*, and *farm*. We then selected three LLMs: DeepSeek (DeepSeek-AI, 2024), Gemma (DeepMind, 2024), and Llama (AI, 2024). At each iteration, a random cluster from these five was selected, and a prompt of the form *"Describe a scene containing a [cluster]."* was designed as the input to the LLM. In Figure 6, you can see the performance comparison and selection ratios of these models.

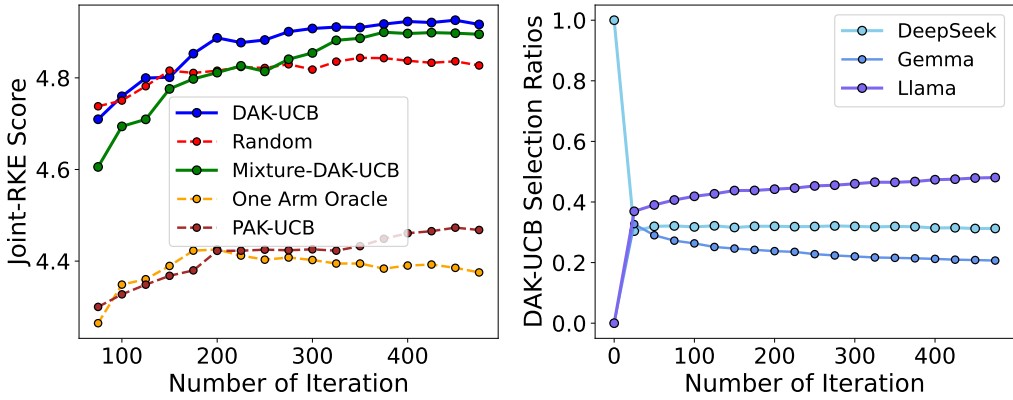

Figure 6: Results are averaged over 10 independent trials.

**Detecting Bias in LLMs Using the I-JRKE Diversity Metric:** In this experiment, we used (DeepSeek-AI, 2024) to generate sentences about cities in the US, Canada, China, and England. One arm was biased toward the capitals of countries, while the other arm was unconstrained. As shown in Figure 7, our algorithm DAK-UCB preferred the unbiased arm, as it resulted in greater diversity across the arms. The results are reported after 200 iterations.

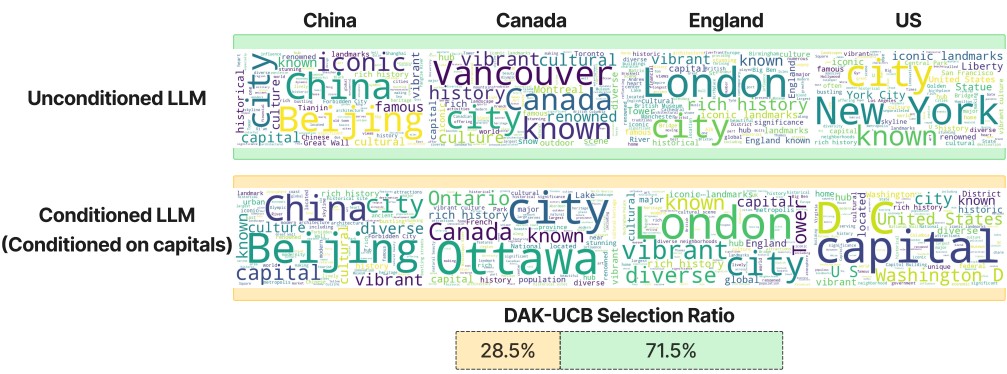

Figure 7: DAK-UCB selection ratio for LLM bias detection experiment.

**Enhancing LLM Diversity via I-JRKE:** In our experiment, we implemented a four-arm setup using DeepSeek, where each arm was diversity-collapsed by country-specific biasing through modified prompts of the form: "Describe one of the famous cities in [Country] in no more than 20 words." (with Japan, France, Brazil, and Egypt as respective biases). Results from 10 runs demonstrated that mixture effectively enhanced diversity. Scores and selection ratios of each arm are presented in Figure 8. The observed decay in DAK-UCB is explainable: With only one cluster, the algorithm sticks to one arm after some point. This naturally converges to a single-arm oracle scenario.

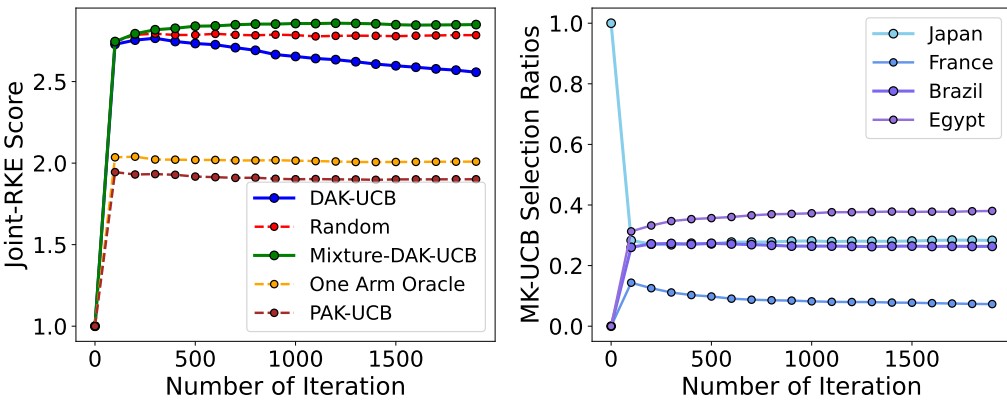

Figure 8: Selection Ratios of biased arms and performance comparison on Joint-RKE.

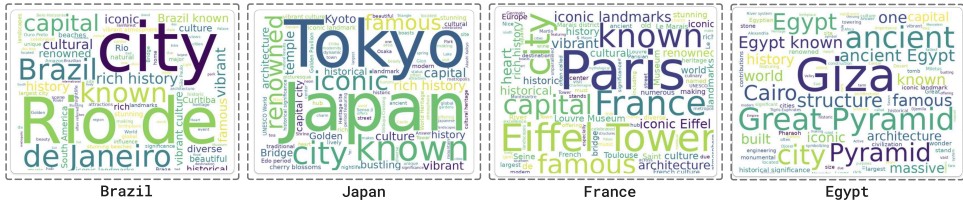

Figure 9: Word Clouds of DeepSeek Responses Across Countries

## C.2 DAK-UCB APPLIED TO IMAGE-CAPTIONING MODEL SELECTION TASK

**Improving Correctness in Image Captioning via JKD:** We evaluated three state-of-the-art image captioning models as our arms: LLAVA (Liu et al., 2023), INSTRUCTBLIP (Dong et al., 2023), and BLIP-2 (Li et al., 2023). At each iteration, we sampled an image-caption pair from a thousand MS-COCO instances, distributed across 10 previously mentioned clusters. By minimizing the Joint Kernel Distance (JKD) between generated captions and MS-COCO reference captions, we aimed to enhance captioning correctness through dynamic model selection. Figure 10 demonstrates the evolution of KID scores across iterations compared to baseline methods, along with the empirical selection ratios for each captioning model. Figure 11 visualizes the dataset.

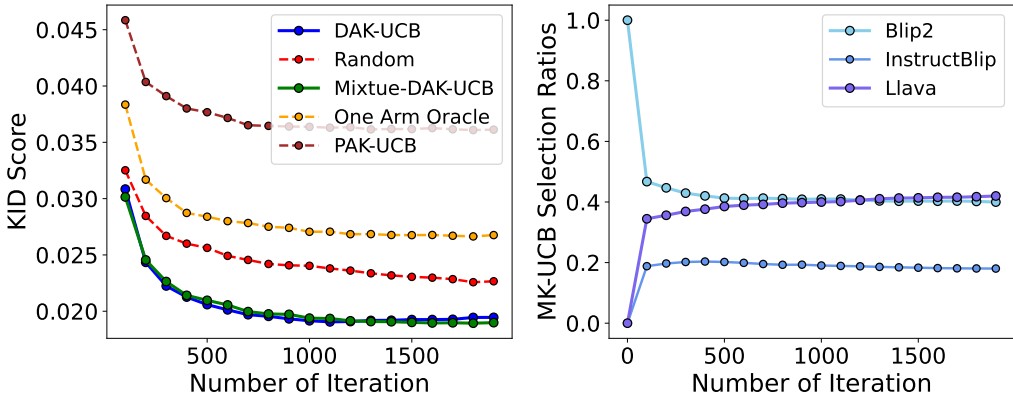

Figure 10: Selection Ratios of image captioning models and performance comparison on KID using JKD metric. Results are averaged over 10 independent trials.

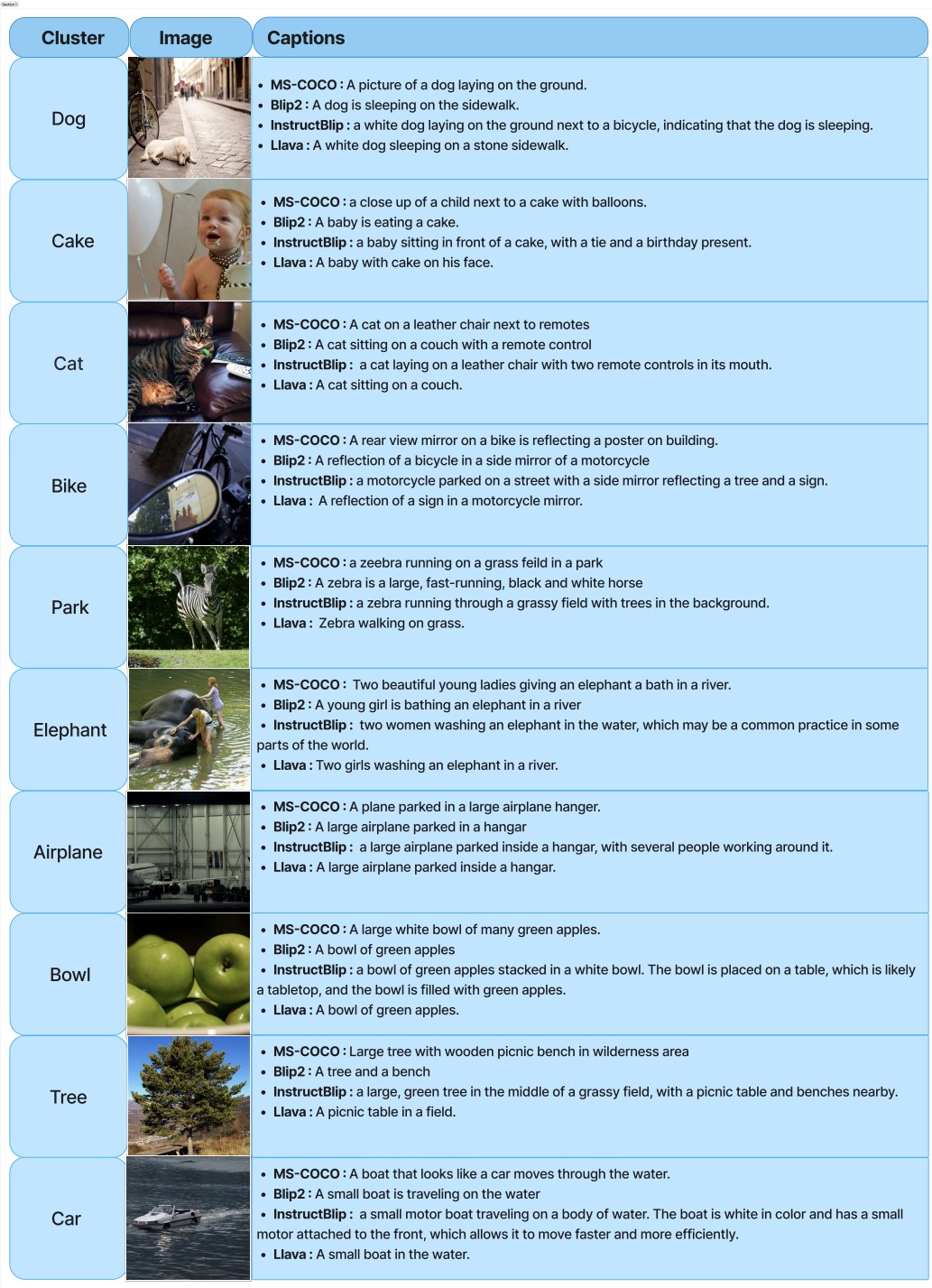

Figure 11: Dataset visualization for image captioning experiment.

**Improving Diversity in Image Captioning via I-JRKE:** We repeated the exact same experiment on image captioning, replacing the JKD objective with I-JRKE to observe if our algorithm can enhance diversity. As shown in Figure 12, our algorithm demonstrated superior performance compared to the baselines. While LLaVA worked best in terms of correctness, our algorithm tends to prefer InstructBLIP for diversity.

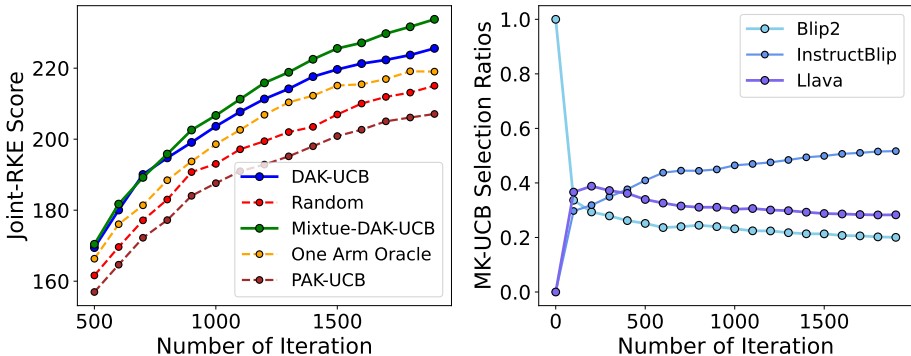

Figure 12: Selection Ratios of image captioning models and performance comparison on Joint-RKE using I-JRKE metric.

**Conditional Expert Selection via I-JRKE Using Classifier-Free Guidance:** We used the dataset from (Rezaei et al., 2025). The dataset was generated by selecting four categories: dog, river, airplane, and building. For each category, GPT was asked to generate 10 adjectives, 10 activities, and 10 places. By mixing these with the category, 1000 prompts were obtained. Some samples of the dataset can be seen in Figure 13. We used SDXL to generate images with classifier-free guidance scales of 2 and 30 for each cluster. We designed the arms as follows: ARM1: Generates images of dogs and rivers with a classifier-free guidance scale of 2.0 (less guided, more diverse) and images of airplanes and buildings with a classifier-free guidance scale of 30.0 (more guided, less diverse). Thus, this arm acts as an expert in the dog and river clusters. ARM2: Does the opposite, making it an expert in the building and airplane clusters. In this experiment, our objective was solely to minimize I-JRKE, and CLIP was not involved in the optimization. As shown in Figure 14 (averaged over 10 trials), the expert for each cluster was successfully detected. However, we observed that in some clusters, the selection ratios were very close. This is acceptable, as we know that a mixture can enhance diversity.

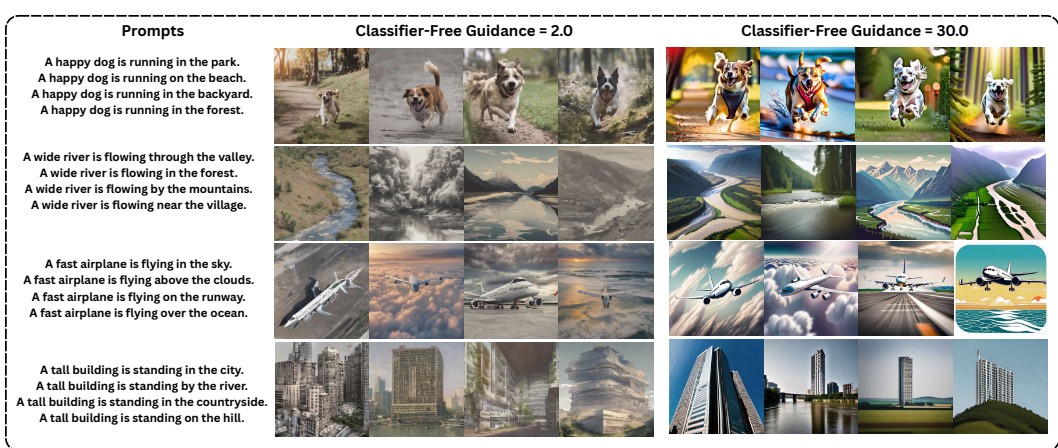

Figure 13: Illustrative examples from the GPT-Generated prompt dataset and their SDXL-generated visualizations under varying guidance scales.

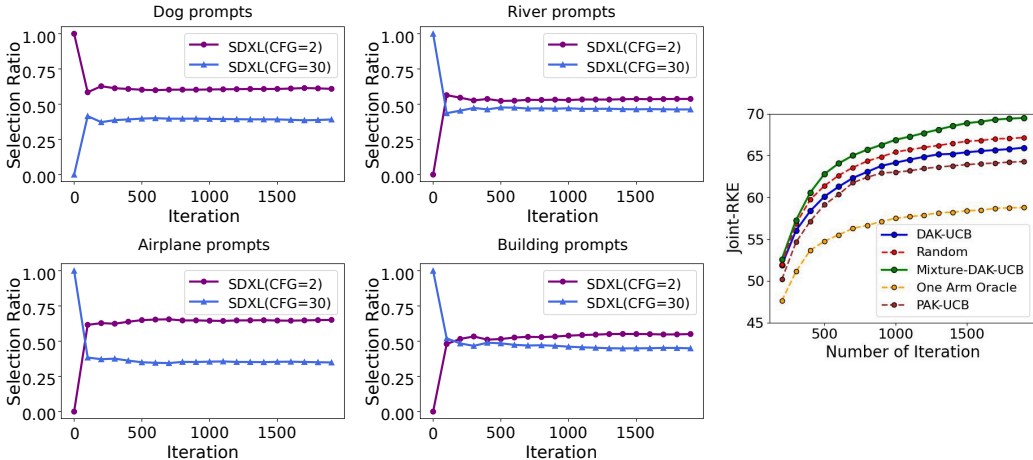

Figure 14: Cluster-conditioned expert selection ratios and performance comparison against baseline methods.

**Diversity Identification in Text-to-Image Generative Models: An Offline Study on Varying Diversity in "animal" Generation:** In this experiment, we modified the setup described in Figure 3 and considered an offline setting. To achieve this, we evaluated the reward with respect to all available offline data and removed the UCB radius term, since the entire dataset was accessible and no exploration was required. Figure 1 presents sample prompts along with the arm-selection preferences of DAK-UCB and Mixture-DAK-UCB.

Table 1: DAK-UCB and Mixture-DAK-UCB Arm Preferences for Sample Prompts

| Prompt | DAK-UCB Selected Arm | Mixture-DAK-UCB Preference Vector |
|---|---|---|
| "an animal in the garden" | Arm 3 (unconditioned) | [0.218, 0.063, **0.719**] |
| "an animal in the meadow" | Arm 3 (unconditioned) | [0.218, 0.064, **0.718**] |
| "an animal near the lake" | Arm 3 (unconditioned) | [0.225, 0.069, **0.706**] |
| "an animal in the jungle" | Arm 3 (unconditioned) | [0.230, 0.082, **0.688**] |
| "an animal in the desert" | Arm 3 (unconditioned) | [0.226, 0.092, **0.682**] |

**Conditional Expert Selection via I-JRKE Using Synthetic Diversity Control:** In this experiment, we utilize four clusters from the MS-COCO dataset: *Bike*, *Car*, *Bowl*, and *Airplane*, each comprising a hundred prompts. The prompts are partitioned into ten groups via K-means clustering applied to their CLIP-embedded vector representations. We designate four specialized experts corresponding to each cluster. When the selected expert matches the revealed prompt's cluster, it generates the appropriate SDXL output for that prompt. For mismatched cases, we employ a diversity-limiting strategy by outputting the image generated for a group representative prompt rather than the actual prompt. Specifically, we assign a representative prompt for each of the ten groups, and for any prompt within a group, the system outputs the SDXL-generated image corresponding to the group's representative. This experiment was conducted over 2000 iterations across 20 independent runs. The resulting expert selection ratios and Joint-RKE scores are presented in Figure 15, The observation that PAK-UCB works competitively well is that expert selection for individual prompts—where each image is generated specifically for its prompt rather than for group representatives—improves CLIP score alongside diversity.

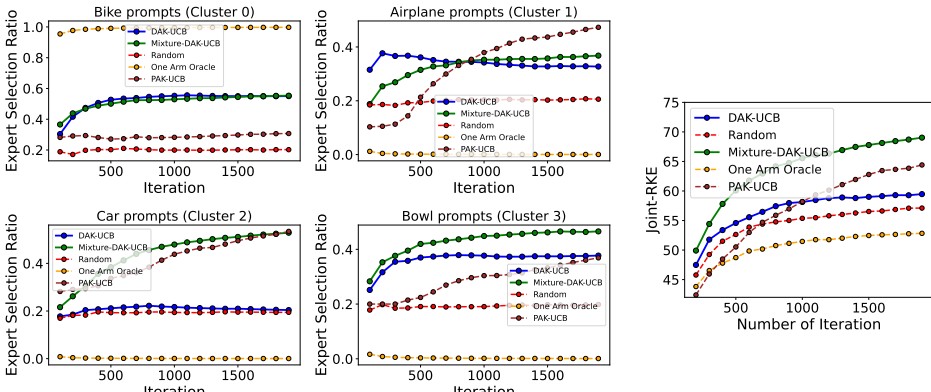

Figure 15: Expert Selection Ratio and Performance Comparison on Joint-RKE.

**Performance Comparison by KID via JKD:** In this experiment, we used the same ten clusters of prompts and three generative models as in the first experiment. We employed MS-COCO images as reference and optimized solely on the JKD score defined earlier. The performance was evaluated using KID to measure how close each baseline gets to the reference. The results are shown in Figure 16. We observed that Mixtue-DAK-UCB and DAK-UCB achieve close performance in this setup, which suggests the optimal solution per cluster is not a mixture.

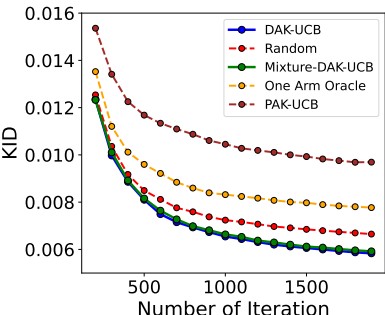

Figure 16: Performance comparison on KID for MS-COCO prompt clusters using Kandinsky, SDXL, and GigaGAN. Results are averaged over 10 trials.

## C.4 ABLATION STUDIES

**Testing the Robustness of Fidelity and Diversity Scores Under Noise:** In this experiment, our objective was to evaluate how reliably our fidelity and diversity metrics behave when the input images undergo controlled degradation. Using 1000 samples from the MS-COCO validation set, we progressively increased the level of blur applied to the images and measured both CLIP-Score fidelity and DINOv2-based diversity. The CLIP-Score decreases monotonically with increased corruption, demonstrating its sensitivity to image quality. Likewise, the diversity metric also decreases, indicating that it does not mistakenly interpret noise as meaningful variation. The two plots in Fig. 17 summarize these trends.

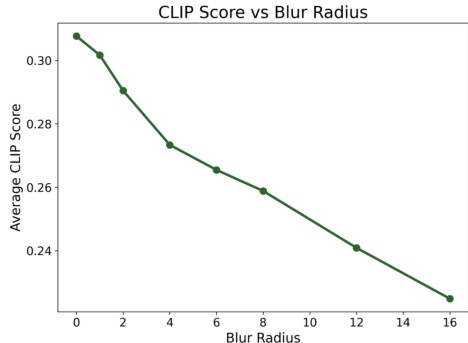 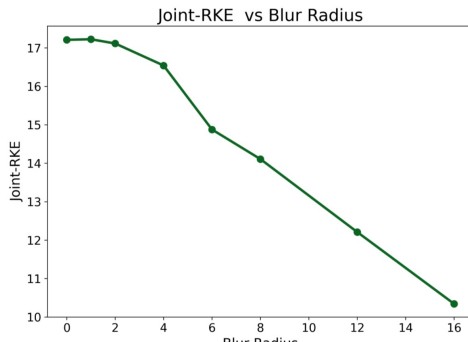

Figure 17: Left: CLIP-Score vs. noise level showing a monotonic fidelity drop. Right: Diversity in DINOv2 embedding vs. noise level showing reduced diversity under stronger corruption.

**Fidelity-Aware Behavior in Asymmetric Degradation:** To verify that our algorithm does not blindly prioritize diversity at the cost of fidelity, we revisited the introductory two-arm experiment shown in Fig. 1. This time, however, we introduced a strong blur (radius = 20) exclusively to the diverse arm while keeping the limited-diversity arm intact. As expected, the selection ratio of the degraded arm dropped sharply, demonstrating that our method correctly down-weights samples whose fidelity deteriorates, even if they originate from a high-diversity source. The final selection ratios are reported in Table 2, showing behavior consistent with our fidelity-sensitive design.

Table 2: Final selection ratios when only the diverse arm is blurred (Radius = 20).

|  | Diverse Arm (Blurred) | Limited-Diversity Arm |
|---|---|---|
| Selection Ratio | 45.32% | 54.68% |

**Efficiency of Random Fourier Feature Approximation:** To address the quadratic growth of kernel computations in DAK-UCB, we adopt Random Fourier Features (RFFs) to approximate the RBF kernel, reducing the per-round computational cost to $O(d^2 t)$ for an RFF dimension $d$. Since the proxy embedding produced by RFFs is $d$-dimensional, this approximation offers a scalable alternative while preserving the behavior of the original kernelized method. In the experimental setting of Fig. 2, we verified that RFF-based DAK-UCB closely matches the performance of the exact RBF-kernel version across all metrics. As shown in Table 3, increasing the number of random features improves stability while maintaining nearly identical scores.

Table 3: Performance of RFF-based DAK-UCB compared to RBF-kernel DAK-UCB under the setup of Fig. 2.

| # RFF Features | Final Joint-RKE | Final CLIP | Final KD ($\times 10^{-3}$) | Elapsed Time ($\times 10^3$) |
|---|---|---|---|---|
| 32 | 158.4 | 29.80 | 7.20 | 4.947 |
| 64 | 160.5 | 30.40 | 6.60 | 4.994 |
| 128 | 162.3 | 31.00 | 5.80 | 5.006 |
| 256 | 172.65 | 31.59 | 4.78 | 5.013 |
| 512 | 182.79 | 32.17 | 3.60 | 5.019 |

**Adaptation to the Introduction of a New Arm:** We repeated the experiment shown in Figure 3 with a modified setup in order to evaluate the algorithm's ability to adapt when a new generative model is introduced partway through the process. Specifically, we began with two arms and introduced the third (and most diverse) arm at iteration 125. Once the new arm appeared, the DAK-UCB algorithm gradually adjusted its selection behavior. As shown in Figure 18, the algorithm successfully adapted to the presence of the newly added arm, redistributing selection ratios and eventually converging to the appropriate mixture for this expanded arm set.

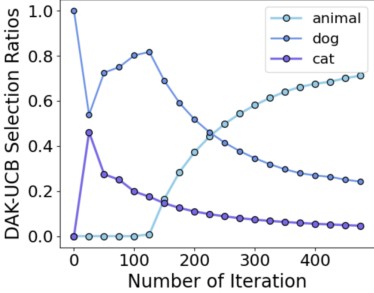

Figure 18: Adaptation behavior after introducing the third (diverse) arm at iteration 125. The algorithm subsequently converges to updated selection ratios that incorporate the new arm.

**Sensitivity to Kernel Function and Embedding Choice:** We repeated the initial experiment (results shown in Figure 2) with a slight modification: we used CLIP embeddings for the images and employed cosine similarity as the kernel function. Figure 19 presents a comparison to the baselines in terms of Joint-RKE and CLIP-score, while Table 4 reports the final cond-vendi scores.

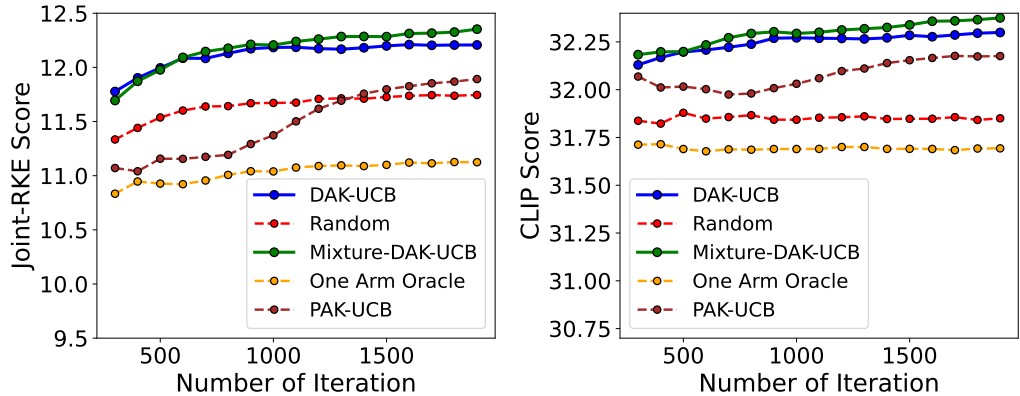

Figure 19: The average CLIP Score and Joint-RKE Diversity score over 10 trials using cosine similarity as kernel function.

Table 4: Comparison of Conditional Vendi scores for algorithms under kernel function sensitivity test.

| Algorithm | Conditional Vendi Score |
|---|---|
| DAK-UCB | 9.92 |
| Mixture-DAK-UCB | 12.53 |
| PAK-UCB | 8.75 |
| Random | 11.82 |
| One-Arm Oracle | 7.99 |

**Sensitivity to the Diversity Term Scalar Hyperparameter:** In this experiment, we used the models from (PixArt-alpha, 2024) and (Runway-ML, 2023) to generate images of an athlete. We generated a hundred images from each model. As suggested by the CLIP, RKE, and Vendi scores in Figure 20, PixArt generated images with higher fidelity, while Stable Diffusion produced more diverse images. To determine if our algorithm can detect this trade-off and to monitor the selection ratios and final metrics for diversity and fidelity under different $\lambda$ values (the diversity term multiplier), we report the results in Tables 5 and 6.

Table 5: Model selection ratio for different $\lambda$ hyperparameter values in Mixture-DAK-UCB algorithm.

|  | $\lambda = 0.0$ | $\lambda = 0.1$ | $\lambda = 1.0$ | $\lambda = 10.0$ |
|---|---|---|---|---|
| PixArt | 65% | 43% | 19% | 12% |
| Stable Diffusion | 35% | 57% | 81% | 88% |

Table 6: Performance metrics for different $\lambda$ hyperparameter values in Mixture-DAK-UCB algorithm.

| Metric | $\lambda = 0.0$ | $\lambda = 0.1$ | $\lambda = 1.0$ | $\lambda = 10.0$ |
|---|---|---|---|---|
| Vendi Score | 15.25 | 22.41 | 33.39 | 35.04 |
| RKE Score | 4.03 | 6.07 | 9.14 | 9.39 |
| CLIP Score | 26.8 | 26.25 | 25.89 | 25.55 |

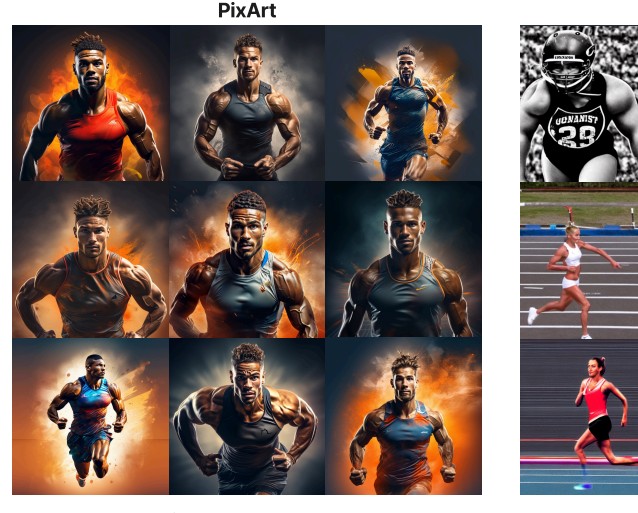

Figure 20: Clip, Vendi, and RKE scores for 100 "image of an athlete" generations, comparing PixArt and Stable Diffusion.

As we increased $\lambda$, the algorithm preferred Stable Diffusion to maximize diversity metrics, which resulted in a corresponding decrease in fidelity metrics. This trade-off is clearly observable in our results. We also observed a gender bias: PixArt tended to generate male athletes, while Stable Diffusion tended to generate female athletes. Our algorithm accounted for gender fairness while enhancing diversity.

**Visualization of Prompt Correlations:**

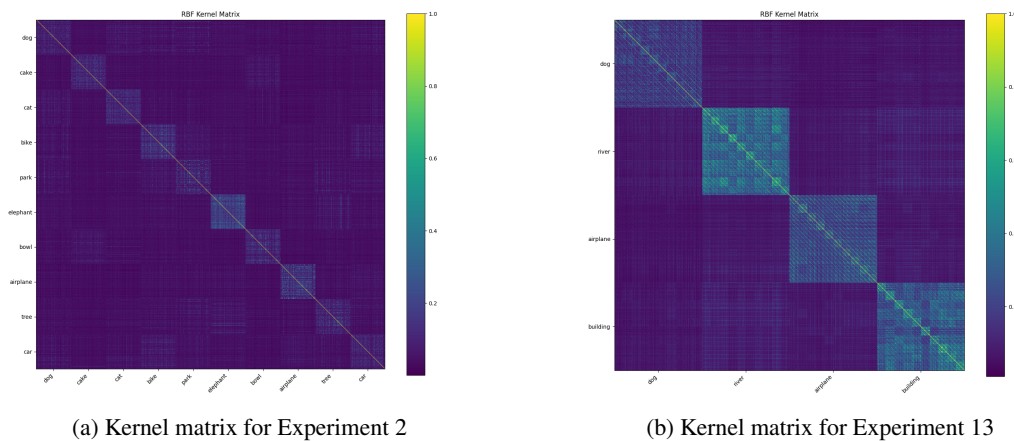

(a) Kernel matrix for Experiment 2          (b) Kernel matrix for Experiment 13

Figure 21: Correlation among prompts in datasets used in Experiments.

## APPENDIX D    STATEMENT ON THE USE OF LARGE LANGUAGE MODELS (LLMS)

LLMs were used solely for proofreading and polishing the language of this manuscript, as well as for generating the prompts of the numerical experiments for the prompt-aware model selection task. All technical content was developed entirely by the authors.

