# OpenReview forum: "DAK-UCB: Diversity-Aware Prompt Routing for LLMs and Generative Models"
_ICLR.cc/2026/Conference — ICLR 2026 Poster_

### Official Review · Reviewer_E2Mq · 2025-10-21

**Soundness:** 3
**Presentation:** 3
**Contribution:** 2
**Rating:** 6
**Confidence:** 3

**Summary:**

The authors propose DAK-UCB a diversity-aware bandit algorithm for selecting the best generative model for a sequence of prompts. The main contribution lies in adding diversity scores to the bandit as well as allowing for a mixture of generative models, instead of selecting a single model per prompt. The experiments show that, compared to simple baselines, the proposed approach 1) achieves better diversity-aware scores, 2) is able to discover the most diverse generative model and 3) does not fall for the trap of selecting generative models that increase diversity in images but based on irrelevant prompts.

**Strengths:**

- Using UCB or a mixture UCB for the selection of a generative model is not new. However, the authors add diversity scores as an additional criterion, which is an original idea. Going beyond the images themselves and incorporating further available information is important in making a more informed choice between generative models.
- The work is grounded in a thorough mathematical argument that is worth sharing with the community and may prompt further research.
- The authors conduct many experiments trying to showcase the strengths of their method in various different contexts.
- The experimental results look promising: Making UCB diversity-aware appears to work as expected, both on (conditional) image generation and text / caption generation tasks.
- The appendix includes ablation studies on the sensitivity to hyperparameter choices, such as the Kernel function.

**Weaknesses:**

- The motivation is weak. For example, to get diverse data with respect to gender, I can just define an appropriate prior distribution of prompts. Hence, the practical benefit over existing models is not always clear.
- $D_g$ is defined twice (in Equation 8 and in line 312 with different notation). $s_g$ is defined twice as well (in Equation 8 and in line 310 with different notation). In general this part is rather inconsistent in notation. Similarly, first $t$ is defined as a text prompt (line 186) and later as a round (line 309).
- I would have expected a comparison to the already existing, and cited, mixture UCB to show the value of adding diversity scores.
- The presented metrics in Figure 2 are the same ones used for the optimization. Therefore, it is no surprise that the proposed models perform best here. Possibly using additional metrics that are not directly part of the algorithm could strengthen the argument.
- Some implementation details remain unclear. For example, after equation 6 and 7 the prompt $t'$ is suddenly no longer considered.
- The method balances fidelity and diversity but the costs in terms of fidelity have not been directly addressed, e.g., there is no CLIP Score in Figure 2.
- The actual costs associated with using this approach for increasing diversity remain unclear. No running times for the experiments are mentioned.  I expect the exploration phase to incur monetary costs that may not be trivial, depending the number of generative models, the complexity of the prompt, etc. While regret bounds are provided, the actual incurred costs for the experiments would be interesting.

**Questions:**

- I am not convinced by the practical use case of the method. If we are interested in diversity along a certain dimension, could we not specify an appropriate prior distribution of prompts, e.g., a 50/50 gender distribution, use existing algorithms to select the most appropriate generative model, and then conditionally sample 50% male and 50% female images? What would be the limits of this approach?
- When do I need to rerun the algorithm? For instance, is a re-training required whenever the user prompt or prompt distribution changes? Could any information from the previous model be salvaged for a warm start?
- In Equation 6 and 7, $t'$ is just another prompt drawn from $P_t$. Should this step not appear in the algorithm?
- In Figure 3, it seems unnatural to have models conditioned on cat or dog classes but keep the prompt about a general "animal". This seems rather detached from how the conditioning is applied in practice. Would you expect different results when being more specific in the prompt?
- In Figure 4, why does the mixture DAK-UCB lead to a worse CLIP score than DAK-UCB, should the former not be nested in the latter?
- How to interpret the expert selection ratio in Figure 4? Should we not expect to select each cluster 25% of the time (assuming the cat, dog, car and cake clusters have equal size)?
- Can the model be used in other domains such as protein generation, where diversity has a clear benefit?

---

> ### Author Response · Authors · 2025-11-28
>
> We sincerely thank reviewer E2Mq for the thoughtful feedback and constructive suggestions on our work. We are pleased that the reviewer found our work “grounded in a thorough mathematical argument” and that our “experimental results look promising”. Regarding the comments and questions in the review,
>
> **1- Motivation of our study**
>
> We appreciate the reviewer’s comment that, when the relevant diversity factors are **known** and **limited in number**, one could attempt to enforce diversity through prompt engineering or factor-specific model selection. Our intention in Figure 1, however, was not to claim that such approaches are impossible, rather it was to illustrate an example of a concrete failure mode of **existing diversity-unaware** model selection methods.
>
> In practice, diversity evaluation in generative modeling often involves **pre-unknown** or **high-dimensional** factors. For example, even in human-centric generation, the number of potential factors of variation (age, gender, race, facial attributes, hairstyle, pose, lighting, etc.) is large, and more generally in non-human generation, the set of relevant factors can be significantly broader and harder to enumerate. As a result, relying on manual, length-bounded prompt engineering becomes more challenging.
>
> This motivates the diversity-aware model selection studied in our work. We highlight that the DAK-UCB method operates effectively even when the diversity factors are not specified beforehand or are too numerous to control manually by prompt engineering. By leveraging prompt-aware model mixtures and evaluating diversity implicitly through coverage-based criteria, the method can capture variation across both known and unknown factors in the available generative models.
>
> **2- Comparison to Mixture-UCB baseline**
>
> Before addressing the reviewer’s question, we would like to clarify the key difference between the baseline Mixture-UCB and our proposed Mixture-DAK-UCB. The Mixture-UCB approach is proposed for online selection of **unconditional (prompt-free)** generative models. Therefore, the application of Mixture-UCB to prompt-based model selection will adopt **the same mixture weights** for model selection of **all input prompts**.
>
> However, in practice, an optimal prompt assignment often requires assigning different prompts to different models, i.e. choosing different mixture weights for different input prompts. This is made possible in the contextual bandit of Mixture-DAK-UCB. In other words, our Mixture-DAK-UCB model selection is both prompt-aware and diversity-aware, whereas the baseline Mixture-UCB is only diversity-aware and not sensitive to the input prompt.
>
> To further demonstrate our point, we added this algorithm as a baseline to the setup used in the experiment reported in Fig. 17, where each arm exhibited diversity collapse in all clusters except one. Below, we report the final Joint-RKE scores for all baselines. As shown, the performance gap between Mixture-DAK-UCB and Mixture-UCB is substantial.
>
> | Method                | Final Joint-RKE Score |
> |-----------------------|---------------------|
> | Mixture-DAK-UCB       | 69.03               |
> | Mixture-UCB           | 65.71               |
> | PAK-UCB               | 64.41               |
> | DAK-UCB               | 59.47               |
> | Random                | 57.11            |
> | One Arm Oracle        | 52.83               |
>
> **3-Definition of $D_g,s_g$ and consistency of math presentation**
>
> We thank the reviewer for pointing out the repeated mentions of the $D_g,s_g$ definitions, and the needed changes for consistency of notations. The updates have been addressed in the revised draft.
>
> **4-Presented metrics in Figure 2**
>
> We would like to clarify that the kernel distance (KD) used to evaluate the algorithm’s performance in Figure 2 differs from the J-RKE metric used to formulate and apply DAK-UCB in the experiment. We note that we included the standard KD score (in DINOv2 embedding) to evaluate the correctness of image generation with respect to the distribution of reference samples. Following the reviewer’s comment, we have also included the plot of CLIP-Scores of the generated images, whose trends similarly support the proposed DAK-UCB algorithm in comparison to the baseline methods.

---

> > ### Author Response · Authors · 2025-11-28
> >
> > **5- Duration of the Exploration Phase of DAK-UCB**
> >
> > We thank the reviewer for the insightful question. Concerning the exploration time and convergence rate, we have now added the selection-ratio plot to Figure 3 to make the convergence behavior explicit. In addition, the final experiment in the main text also reports arm-selection ratios. These results together illustrate that the convergence rate naturally depends on the complexity of the underlying dataset. In the synthetic setting of Figure 2, where the structure is simpler, the algorithm converges rapidly. In contrast, in the final experiment based on the more complex and diverse MS-COCO dataset, the convergence is slower, as expected.
> >
> > **6-”When do I need to rerun the algorithm?”**
> >
> > Regarding this question, we highlight that an advantage of the online learning DAK-UCB method is its adaptivity to changes in the arms’ distributions, i.e., it does *not* require restarting or retraining the algorithm when the user prompt or prompt distribution changes. To test this, we evaluated the method in the following scenario where a *new arm* (i.e., a new generative model) is introduced partway through the iterations: We repeated the experiment we had on generating animal images in Figure 3 and introduced the most diverse arm at iteration 125. Below, we report how the algorithm detected the diversity of this arm and the corresponding selection ratios across different iterations.
> >
> > | Arm    | Iteration 125 | Iteration 250 | Iteration 375 | Iteration 500 |
> > |--------|----------------|----------------|----------------|----------------|
> > | Dog    | 82%            | 42%            | 28%            | 24%            |
> > | Cat    | 18%            | 2%             | 1%             | 1%             |
> > | Animal | 0%             | 56%            | 71%            | 75%            |
> > ---
> >
> >
> > **7- Clarification on variable $t’$**
> >
> > We would like to clarify that in the definition of the underlying score, $t’$ represents the random variable of the prompt which is i.i.d. sampled with the prompt random variable $t$. Note that in the implementation of DAK-UCB with the empirical distribution of observed prompts in the previous rounds, $t'$ corresponds to the prompt variable uniformly sampled from the set of previously observed prompts.
> >
> > **8- Clarifications on Figure 3’s experiment**
> >
> > We would like to clarify that in Figure 3 the arms only have knowledge conditioned on the *general* prompt and do not have prior information about diversity within each specific animal subclass (e.g., cats vs. dogs). Consequently, when the context becomes more specific, the algorithm has to *learn the diversity structure of this new context from scratch*, and its short-term behavior may not be immediately predictable. We will make this clearer in the text.
> >
> > **9- CLIP-Score Comparison of DAK-UCB and Mixture-DAK-UCB in Figure 4**
> >
> > We would like to clarify that the improvement of Mixture-DAK-UCB over (non-mixture) DAK-UCB is in terms of distributional correctness that is captured by the Kernel Distance. Please note the KD score aggregates diversity and fidelity of the generated data. While the Mixture-based algorithm improves the diversity score, it may not result in an improvement of a pure fidelity score, such as the CLIP-Score. In Figure 4, we also observed that our proposed DAK-UCB can reach a slightly higher average CLIP-Score, yet we emphasize that the KD-based distributional correctness (comprising both fidelity and diversity) is better for the Mixture-DAK-UCB. We will explain this point more clearly in the revision.
> >
> >
> > **10-Clarification on selection ratios in Figure 4**
> >
> > We clarify that we have measured the selection ratios with respect to prompt population in each of the four equally-sized prompt categories. Therefore, to obtain the selection ratio with respect to the entire prompt set, one needs to multiply the reported numbers by 4.
> >
> >
> > **11- Application of DAK-UCB to non-image domains**
> >
> > We thank the reviewer for the suggestion. We highlight that we have already presented the application of our proposed DAK-UCB to multi-LLM prompt assignment in Appendix C.1 and image captioning models in Appendix C.2. Extending the application of DAK-UCB to protein and graph generative models will be relevant future directions, which we will discuss in the revised conclusion.

---

### Official Review · Reviewer_a2p3 · 2025-11-01

**Soundness:** 3
**Presentation:** 2
**Contribution:** 3
**Rating:** 6
**Confidence:** 2

**Summary:**

The paper proposes DAK-UCB, a contextual-bandit algorithm for the online selection of generative models with diversity considerations. It incorporates fidelity and diversity-related metrics into the selection process. The experimental results demonstrate considerable improvements.

**Strengths:**

1) The topic of this paper is meaningful. The new method of delicate online model selection is valuable for the community.
2) The experimental results on benchmarks demonstrate considerable improvements.

**Weaknesses:**

1) I do wander the cost of Mixture-DAK-UCB: it solves quadratic program each round and stores matrices, will this bring scalability concerns?
2) More details and analysis of the bias evaluation beyond gender is necessary.

**Questions:**

See the weakness.

---

> ### Author Response · Authors · 2025-11-28
>
> We sincerely thank reviewer a2p3 for the thoughtful feedback and suggestions on our work. We are glad that the reviewer found the topic of our work “meaningful” and the method for online model selection “valuable for the community”. Regarding the comments and questions in the review,
>
> **1-Time Complexity of Mixture-DAK-UCB**
>
> We thank the reviewer for the insightful point. As mentioned in the work, the number of kernelized regression tasks in Mixture-DAK-UCB scales with $m^2$, the square of the number of models $m$. This is while the number of kernelized regression tasks for $m$ models is $m$. However, please note that the resulting optimization problem is a quadratic convex optimization that can be efficiently solved by convex optimization solvers such as CVXPY.
>
> To assess the extra time complexity introduced by the mixture formulation of Mixture-DAK-UCB, we run DAK-UCB with and without the mixture component using the same setup as in the first numerical experiment (Fig. 2). As shown in the following table, the trajectories are very close, indicating that the additional computational cost introduced by the mixture is negligible in practice.
>
> | Algorithm         | 500 iters (s) | 1000 iters (s) | 1500 iters (s) | 2000 iters (s) |
> |-------------------|-----------------------|-------------------------|-------------------------|-------------------|
> | DAK-UCB           | 605.65     | 1631.91      | 3082.88      | 4999.66       |
> | Mixture-DAK-UCB   | 608.88      | 1649.41      | 3131.83      | 5046.95       |
>
> **2-Beyond gender diversity evaluation**
>
> To address the comment and further demonstrate the algorithm’s application where the diversity is not gender-related, we include two additional scenarios: a simulated text-to-image setting where the arms exhibit biases toward specific animals (in the paper’s Fig. 3), and a real-world LLM setting where different models are biased toward particular celebrities or cities. In both cases, the algorithm successfully identifies these biases and adjusts its selection accordingly.
>
> These results support that the proposed method is not limited to a single type of bias or modality. Instead, it consistently detects and responds to diverse forms of changes across different domains, supporting our claim that the algorithm can reliably observe and understand underlying biases in the arms.
>
> To provide a concrete illustration of these points, we also evaluated our method on three widely used open-source LLMs (Llama3.2, Qwen2, and Gemma3 in Ollama repository). At each iteration, the models were prompted with:  “A sentence about a vibrant city in Northern America.” or “A sentence about a renowned celebrity”
>
> Across all prompt types, we observed that each model produced sentences with *very limited geographical or celebrity-character diversity*, but the few captured modes differed across the tested LLMs. For the “City in Northern America” sentence in particular, Llama3.2 frequently defaulted to "New Orleans", Gemma3 repeatedly generated sentences about "Chicago/New York", and Qwen2 consistently produced outputs about "New York".  For the queried “Celebrity” sentence, Llama3.2 frequently defaulted to "Leonardo DiCaprio/Taylor Swift/Meryl Streep", Gemma3 consistently generated sentences about "Taylor Swift", and Qwen2 consistently produced outputs about "Elon Musk/Meryl Streep". These complementary biases demonstrate that while each individual model collapses to a single dominant city or celebrity, they do so in **different** ways.
>
> The Mixture-DAK-UCB algorithm leverages these complementary failure modes. Rather than inheriting the collapse of a single model. As a result, the selected mixture achieves substantially higher qualitative and quantitative diversity. This is reflected in the conditional Vendi scores and mixture weights reported below:
>
> >"A vibrant city in Northern America"
>
> | Model        | Vendi Score      | Percentage                                      |
> |--------------|------------------------|------------------------------------------|
> | Llama3.2        | 1.72 ± 0.02            | 100% Llama – 0% Gemma – 0% Qwen2                 |
> | Gemma3        | 2.39 ± 0.03            | 0% Llama – 100% Gemma – 0% Qwen2                 |
> | Qwen2        | 1.45 ± 0.02            | 0% Llama – 0% Gemma – 100% Qwen2                 |
> | **Mixture-DAK-UCB**  | **2.77 ± 0.03**        | **20% Llama – 68% Gemma – 12% Qwen2**            |
>
> >"A renowned celebrity"
>
> | Model        | Vendi Score      | Percentage                                      |
> |--------------|--------------------|--------------------------------|
> | Llama3.2        | 2.25 ± 0.02            | 100% Llama – 0% Gemma – 0% Qwen                 |
> | Gemma3        | 1.14 ± 0.02            | 0% Llama – 100% Gemma – 0% Qwen                 |
> | Qwen2         | 2.52 ± 0.03            | 0% Llama – 0% Gemma – 100% Qwen                 |
> | **Mixture-DAK-UCB**  | **2.87 ± 0.03**        | **33% Llama – 7% Gemma – 60% Qwen**             |

---

### Official Review · Reviewer_rNZi · 2025-11-01

**Soundness:** 2
**Presentation:** 3
**Contribution:** 2
**Rating:** 4
**Confidence:** 3

**Summary:**

This paper recasts prompt‑conditioned model routing as a contextual bandit that balances generation quality and diversity when choosing among generative models. Building on Upper Confidence Bound, which optimizes only fidelity, the proposed DAK‑UCB jointly estimates prompt fidelity and diversity using separate regressors and an optimism bonus. The method also extends to prompt‑conditioned mixture selection that learns routing probabilities. The authors evaluate DAK‑UCB on the MS‑COCO validation set and on a synthetic dataset created with GPT‑4o to assess its effectiveness.

**Strengths:**

1. The paper presents a complete and clearly derived theory. It details how KD and RKE are extended to metrics suitable for joint variables and provides the theoretical basis for a diversity‑aware extension of UCB.
2. DAK‑UCB is applied to image generation, text generation, and image captioning, and the experiments validate the approach across these tasks.

**Weaknesses:**

1. The only major difference from PAK‑UCB is the shift from prompt‑aware selection alone to a joint selection based on both prompt fidelity and diversity. Many other aspects, including the theoretical tools and the choice of backbone models in experiments, overlap substantially, which reduces the perceived novelty.
2. The DAK‑UCB score relies on CLIP text embeddings and DINOv2 image embeddings as the algorithm’s inputs, and the evaluation uses the same embeddings. Prior work indicates that CLIP and DINOv2 can encode biases stemming from imbalances in their training data across geography and race. Using biased embeddings may affect distances between text‑image vector pairs and thereby influence routing, yet the paper does not analyze whether this impacts DAK‑UCB’s outcomes.
3. In this work, fidelity refers only to alignment between an image and its prompt, not to the realism of the image itself. It therefore remains unclear whether DAK‑UCB can select diffusion models that produce more realistic images, for example better FID.

**Questions:**

See “Weakness”

---

> ### Author Response · Authors · 2025-11-28
>
> We thank reviewer rNZi for the thoughtful feedback on our work. We are glad that the reviewer finds our work to present “complete and clearly derived theory”. Regarding the comments and questions in the review,
>
> **1-Novelty with respect to the PAK-UCB baseline**
>
> We kindly refer the reviewer to Item 1 of our global response. Here, we would like to further highlight that the solution of diversity-aware model selection for a *single prompt* could be a **non-degenerate mixture** of the generative models, resulting in our proposal of *Mixture-DAK-UCB*, as the optimal solution to the diversity-aware selection of the models can potentially be a non-degenerate mixture of the arms assigned to an individual prompt. In contrast, in the fidelity-based selection of PAK-UCB, the optimal assignment for each prompt is only one model, because the averaged equality score of a mixture is always bounded by the maximum individual quality score.
>
>
> To provide an illustrative example for these points, we conducted experiments on three standard open-source LLMs (Llama3.2, Qwen2, and Gemma3 in Ollama repository). At each iteration, the prompt was either “A sentence about a vibrant city in Northern America” or “A sentence about a vibrant city in Europe” with equal probabilities. We observed that each model exhibited *limited diversity* in the choice of city, but *in different complementary ways*.  For prompts with “Northern America”, Llama3.2 mentioned “New Orleans” more frequently (Vendi score: 1.72), Gemma3 frequently mentioned “Chicago/New York” (Vendi score: 2.39), and Qwen2 consistently mentioned “New York” (Vendi score: 1.45).
>
> On the other hand, for the prompts with “Europe”, the diversity ranking was different, Llama3.2 mentioned “Barcelona/Amsterdam/Porto” more frequently (Vendi score: 2.18), Gemma3 always mentioned “Paris/Rome” (Vendi score: 1.55), and Qwen2 more frequently mentioned “London/Vienna/Berlin” (Vendi score: 2.33).
>
>
> Thus, although each model had a limited city diversity, maybe concentrating on one or few cities, the chosen Mixture by Mixture-DAK-UCB assigned non-zero weight to each model for every prompt, resulting in higher qualitative and quantitative (Vendi score) diversity. Furthermore, the optimal mixture weights for the prompts are significantly different, motivating the prompt-aware formulation in our proposed DAK-UCB algorithm.
>
>
> >"A vibrant city in Northern America"
>
> | Model       | Vendi Score       | Percentage                             |
> |-------------|--------------------|-----------------------------------------|
> | Llama3.2       | 1.72 ± 0.02        | 100% Llama – 0% Gemma – 0% Qwen2        |
> | Gemma3       | 2.39 ± 0.03        | 0% Llama – 100% Gemma – 0% Qwen2        |
> | Qwen2       | 1.45 ± 0.02        | 0% Llama – 0% Gemma – 100% Qwen2        |
> | **Mixture-DAK-UCB** | **2.77 ± 0.03**    | **20% Llama – 68% Gemma – 12% Qwen2**   |
>
>
> >"A vibrant city in Europe":
>
> | Model       | Cond-Vendi Score    | Percentage                             |
> |-------------|----------------------|-----------------------------------------|
> | Llama3.2      | 2.18 ± 0.02          | 100% Llama – 0% Gemma – 0% Qwen2        |
> | Gemma3       | 1.55 ± 0.02          | 0% Llama – 100% Gemma – 0% Qwen2        |
> | Qwen2       | 2.33 ± 0.02          | 0% Llama – 0% Gemma – 100% Qwen2        |
> | **Mixture-DAK-UCB** | **2.63 ± 0.03**      | **44% Llama – 8% Gemma – 48% Qwen2**    |
>
>
> ----
>
> **2- The choice of embeddings in DAK-UCB and embedding biases**
>
> We would like to clarify that our proposed DAK-UCB method is agnostic to the choice of image and text embeddings, and we do not restrict the algorithm’s application to the specific  CLIP and DINOv2 embeddings. The algorithm can be applied with any choice of embeddings.
>
> We used the CLIP and DINOv2 embeddings in our numerical experiments because they have been recommended in recent literature. The CLIP-Score, proposed in [1], using the CLIP image and text embeddings, is a widely-used metric for text-to-image fidelity evaluation. Also, the choice of DINOv2 for image generation evaluation is recommended by the recent paper [2] on the biases of image embeddings in generative model evaluation.
>
> To further demonstrate the application of our method with open-source embedding models, we redid the experiment of Figure 2, when replacing the CLIP encoder with OpenCLIP (trained on the open-source LAION dataset). The resulting joint-RKE performance comparison is presented below.
>
> |                     | DAK-UCB | Mixture-DAK-UCB | Random  | One Arm Oracle | PAK-UCB |
> |---------------------|---------|------------------|---------|----------------|---------|
> | Final Joint-RKE     | 140.40 | 138.58          | 132.8 | 117.913        | 100.744 |
>
>
> [1] Hessel et al, “CLIPScore: A Reference-free Evaluation Metric for Image Captioning”, EMNLP 2021\
> [2] Stein et al, “Exposing flaws of generative model evaluation metrics and their unfair treatment of diffusion models” NeurIPS 2023

---

> > ### Author Response · Authors · 2025-11-28
> >
> > **3-“In this work, fidelity refers only to alignment between an image and its prompt”**
> >
> > We respectfully clarify that we do not suggest ignoring photo realism in the fidelity evaluation process. To explain this, we would like to make the following clarifications about our theoretical formulation and numerical experiments of DAK-UCB. We will include the clarification in the revised text.
> >
> > Regarding the theoretical formulation of DAK-UCB, we highlight that the fidelity evaluation in DAK-UCB applies to any user-specified sample-level fidelity evaluation score $\phi:\mathcal{T} \times \mathcal{X} \to \mathbb{R}$, i.e. the fidelity score is considered to be the average of the sample-level fidelity score for any arbitrary function $\phi$. Such a sample-level fidelity evaluation can quantify any combination of the prompt-image alignment and image-only quality.
> >
> > Regarding our numerical experiments, we chose the CLIP-Score for sample-level fidelity, as it is a standard and widely-adopted text-to-image fidelity score. We still note that the end-user can change the CLIP-score to any other score and directly apply the algorithm (as we discussed in the previous item).
> >
> > Furthermore, we highlight that the CLIP-Score is not merely a function of image-text alignment; indeed, it is sensitive to photorealism. To demonstrate this, we considered image blurring and evaluated the averaged CLIP-Score over the benchmark MS-COCO validation set as the blurring intensified. The following table shows that the score dropped sharply as the blurring intensified. We will include this table in the revised Appendix.
> >
> > | Blur Radius | 0     | 4     | 8     | 12    | 16    |
> > |-------------|-------|-------|-------|-------|-------|
> > | CLIP Score  | 30.77 | 29.04 | 26.54 | 24.03 | 22.42 |

---

### Official Review · Reviewer_4Gv8 · 2025-11-12

**Soundness:** 2
**Presentation:** 3
**Contribution:** 2
**Rating:** 4
**Confidence:** 3

**Summary:**

The paper studies the online selection of generative models for given prompts, with explicit consideration of both fidelity and diversity in the generated outputs.
The existing prompt-aware selection methods only focus on fidelity (e.g., CLIP-Score), often overlooking diversity, which can lead to repetitive or biased outputs. To overcome this, the authors propose a method, Diversity-Aware Kernelized Upper Confidence Bound (DAK-UCB), that extends the kernelized UCB framework by incorporating diversity metrics
(joint kernel distance (JKD) and joint Rényi kernel entropy (JRKE)) into the selection process to balance fidelity and diversity. The authors also propose a mixture variant (Mixture-DAK-UCB) that allows for prompt-conditioned mixtures of models. The approach is validated on text-to-image and language model generation tasks, showing improvements in both diversity and fidelity metrics compared to baselines.

**Strengths:**

**The following are the strengths of the paper:**
1. This paper considers the online selection of generative models for given prompts that explicitly consider both fidelity and diversity in the generated outputs, overcoming repetitive or biased outputs.

2. The authors propose a new algorithm, Diversity-Aware Kernelized Upper Confidence Bound (DAK-UCB), that incorporates diversity metrics into the selection process to balance fidelity and diversity, while having theoretical guarantees on the regret upper bound (better bound for Sup-DAK-UCB).

3. The authors empirically validated the proposed algorithm on text-to-image and language model generation tasks, showing improvements in both diversity and fidelity metrics compared to baselines.

**Weaknesses:**

**The following are the weaknesses of the paper:**
1. The kernel methods and the estimators may lead to high computational overhead (due to kernel matrices), especially when the number of prompts (or models) increases. There should be more details on how the computational cost varies with the increase in samples (number of iterations)

2. It is unclear why the paper only focuses on JKD and JRKE, as there are other diversity metrics.

3. There is not much novelty in the regret bounds, as the derivation depends on existing results, which make many assumptions (normalized kernels, sub-Gaussian noise, RKHS boundedness), which may not hold in all practical settings, especially for LLMs.

4. The diversity-aware selection may degrade fidelity, and there needs to be a discussion about this trade-off. Furthermore, there should be a discussion about the choice of kernel functions, as it may depend on the prompts and models.

**Questions:**

Please address the weaknesses of the paper.

**Details Of Ethics Concerns:**

I do not find any ethical concerns.

---

> ### Author Response · Authors · 2025-11-28
>
> We sincerely thank Reviewer 4Gv8 for the thoughtful feedback on our work. Regarding the comments and questions in the review,
>
> **1-Computational overhead of the kernelized UCB algorithm in DAK-UCB**
>
> As we have discussed in the text, it is known in the literature that the kernelized-UCB approach leads to a computational overhead of $O(t^3)$ at Round $t$ for solving the kernelized regression problems. The reduction of computational costs of general kernelized-UCB methods has been a topic of active research in the literature, and several ideas have been proposed to control the growth of the computational cost with the iteration round, which are applicable to our proposed DAK-UCB method.
>
> Specifically, to lower the $O(t^3)$-growing computational complexity of the general kernel-based DAK-UCB, one can implement the selection algorithm in the kernel feature space for a finite-dimensional feature map $\phi:\mathcal{X}\to\mathbb{R}^d$ of the target kernel $k(x,y) = \langle \phi(x) ,\phi(y)\rangle$. This is especially the case for the linear and normalized linear kernels leading to the Lin-UCB counterpart of DAK-UCB. The resulting computational cost will be $O(d^2 t)$ linearly growing with iteration $t$.
>
> For shift-invariant kernel functions, including the RBF kernel used in our experiments, we can use the established random Fourier features (RFFs) (Rahimi \& Recht, 2007) to lower the growing computational cost to $O(d^2 t)$ for $d$ random Fourier features (Note that the proxy RFF kernel has a feature map of dimension $d$). We have tested the RFF implementation of DAK-UCB, and in the case of our Figure 2’s experiments, it repeats the performance scores of RBF kernel DAK-UCB as we highlight in the following table:
>
> | # RFF Features | Final Joint-RKE Score | Final CLIP Score | Final KD Score (×10⁻³) | Elapsed Time (s) $\times 10^3$ |
> |----------------|-----------------------:|------------------:|------------------------:|------------------:|
> | 32             | 158.4                 | 29.80             | 7.20                   | 4.947         |
> | 64             | 160.5                 | 30.40             | 6.60                   | 4.994         |
> | 128            | 162.3                 | 31.00             | 5.80                   | 5.006         |
> | 256            | 172.65                | 31.59             | 4.78                   | 5.013         |
> | 512            | 182.79                | 32.17             | 3.60                   | 5.019         |
>
>
> **2-The choice of JRKE and JKD diversity metrics in the DAK-UCB framework**
>
> We kindly refer the reviewer to Items 1 and 2 of our general response.
>
>
> **3-Regret bound in Theorem 1**
>
> As we have discussed in the paper, our goal in Theorem 1 is not to introduce a new approach for regret analysis of kernelized-UCB, which is already well-studied in the online learning literature. Instead, we aim to show that *the standard guarantees can be extended to the diversity-aware objective in DAK-UCB*. Please note that this extension is not trivial: unlike fidelity-based rewards, our suggested diversity metrics depend on pairs of samples and therefore do not admit the per-sample decomposition required by standard kernelized-UCB analyses.
>
> As highlighted in Item 2 of the general response, we specifically show that **JRKE and JKD uniquely admit a two-sample expectation structure** that allows the diversity-aware objective to be rewritten as an expectation over a single-sample reward function, enabling the application of kernelized-UCB confidence bounds. This property is essential for obtaining regret guarantees; using other diversity metrics (Recall, Coverage, Vendi) would not allow the same analysis.

---

> ### Author Response · Authors · 2025-11-28
>
> **4-Diversity Fidelity trade-off in DAK-UCB**
>
> We understand the reviewer’s comment that a diversity-aware selection method may potentially lead to a degradation in the image quality for a higher diversity. However, please note that our evaluation of fidelity and diversity are happening in **CLIP and DINOv2 embedding spaces**, as CLIP-Score measures the fidelity in the CLIP image embedding space and the JKD and JRKE measure diversity in the DINOv2 embedding space.
>
>
> Therefore, any sacrifice in the output image quality to obtain a higher diversity requires the embedding models to count the visually-poor image as a diverse image. Our experimental evaluation indicates that the CLIP and DINOv2 models (which are recommended in the literature for assessing the image generation task [1,2]) are relatively robust to noise in the image, and would assign lower diversity scores to noisy images, i.e., they do not count noise as diversity.
>
>
> To demonstrate this point, we have run the following two experiments and will include the results in the revised manuscript. In the first experiment, we aimed to verify that the CLIP-Score fidelity term will decrease with raising noise intensity added to the input images. We used the MS-COCO validation dataset, applied increasing levels of blurring to the images, and observed a monotonic drop in CLIP scores as in the following table.
>
> | Blur Radius | 0     | 4     | 8     | 12    | 16    |
> |-------------|-------|-------|-------|-------|-------|
> | CLIP Score  | 30.77 | 29.04 | 26.54 | 24.03 | 22.42 |
>
> In the second experiment, we tested whether the diversity metric (in the DINOv2 embedding) would incorrectly count noise as diversity. Using the blurred images of MS-COCO validation set, we measured the averaged diversity score and found that it also decreases as blur radius increases, supporting that our proposed DINOv2-based metric does not count noise in the image as diversity, and indeed will decrease as noise power is increasing.
>
> | Blur Radius | 0     | 4     | 8     | 12    | 16    |
> |-------------|-------|-------|-------|-------|-------|
> | Joint-RKE   | 17.29 | 17.11 | 14.84 | 12.20 | 10.34 |
>
>
> [1] Hessel et al, “CLIPScore: A Reference-free Evaluation Metric for Image Captioning”, EMNLP 2021\
> [2] Stein et al, “Exposing flaws of generative model evaluation metrics and their unfair treatment of diffusion models” NeurIPS 2023

---

### Author Response · Authors · 2025-11-28
**Authors' General Response**

We sincerely thank the reviewers for their constructive feedback and suggestions. We provide point-wise responses under each review textbox. In this general response, we aim to clarify (1) the novelty of DAK-UCB relative to existing contextual bandit algorithms and (2) the motivation for using JRKE and JKD as the diversity and correctness scores in our framework, to address comments raised by Reviewer 4Gv8 and Reviewer rNZi.

**1- Novelty of our framework relative to existing contextual bandit algorithms**

We would like to highlight that standard contextual bandit (CB) methods **cannot address** *the diversity-aware model selection task*, because a diversity-aware selection requires **set-level** evaluation, whereas standard CB algorithms (including that of baseline PAK-UCB method) are designed for *per-sample* rewards.

In a standard CB setting, the learner receives a reward *independently at each round ( i.e., for each generation in the online model selection)*. Mathematically, if the reward for (prompt,output) pair $(t_i, x_i)$ at round $i$ is $\phi\bigl( t_i, x_i\bigr)$, the aggregate reward over $T$ rounds in standard CB is:

$$\frac{1}{T}\sum_{i=1}^T \phi(t_i, x_i)$$

which explicitly assumes that rewards decompose additively across generation rounds.

However, the diversity of generations cannot be decomposed across individual samples at different rounds. The diversity of a generated set of samples depends on their **relative positioning**, which cannot be captured by averaging individually assigned scores. This is why all the diversity metrics such as RKE, Vendi, Recall, or coverage-based measures require evaluating **pairs** or **sets** of samples. Therefore, existing CB algorithms that only receive per-sample rewards cannot capture the diversity behavior of the candidate models.

This gap also explains why the optimal policy in our proposed DAK-UCB can be a **non-degenerate mixture** over models for a single prompt: achieving higher diversity could require drawing from multiple generators. In contrast, in only-fidelity-based selection (e.g., PAK-UCB), the mixture-based model selection cannot outperform the best single model, and hence the optimal solution collapses to a single arm. This structural difference further illustrates why diversity-aware selection falls outside the scope of existing contextual bandit formulations.

To the best of our knowledge, our work is the first to study the diversity-aware CB methods and to propose a CB algorithm to choose a mixture of arms for an input context.

---

**2. Why we suggest JRKE and JKD scores for diversity-aware model selection**

We propose applying the JRKE and JKD scores, because they possess a unique **two-sample expectation form**, where the score reduces to the following form for a function $\kappa:\mathcal{X}\times\mathcal{X}\to \mathbb{R}$ and *independently-drawn* $X,X’$ from $P$:

$$\text{Score}(P) = \mathbb{E}_{X, X' \sim P}[\kappa(X, X')]$$

This lets us rewrite the score as $\mathbb{E}_{X \sim P}[\phi(X)]$ where $\phi(x) = \underset{X' \sim P}{\mathbb{E}}[\kappa(x, X')]$

which gives *sample-level* access to a function $\phi$ whose expectation recovers the true diversity or correctness metric. Importantly, i) $\phi(x)$ can be **estimated efficiently** using Monte Carlo over a few generated samples, ii) $\phi(x)$ has a well-behaved gradient suitable for kernelized-UCB updates, iii) the mean of $\phi(x)$ exactly recovers I-JRKE/JKD.

The other standard diversity measures (including Recall, Coverage, Vendi) **do not reduce** to an expectation of a per-sample function of the above form and therefore cannot be estimated or optimized efficiently in an online contextual bandit setting. The two-sample expectation property of JRKE/JKD is therefore essential for enabling efficient online diversity evaluation and is a central technical reason behind our metric choices.

---

### Author Response · Authors · 2025-12-03
**Wrap-Up: Summary of Clarifications and Manuscript Updates**

We sincerely thank the reviewers for their constructive feedback and suggestions. We have uploaded a revised manuscript with the updates highlighted in blue to improve clarity and presentation in accordance with the points discussed in our responses. Below, we summarize the key clarifications and updates provided during the discussion phase.

**1- Sample Set-Level Diversity Evaluation and the Need for DAK-UCB**

We clarified that diversity-aware model selection fundamentally requires set-level evaluation (pairwise or multi-sample structure). Since standard contextual bandits (including PAK-UCB) optimize only per-sample rewards, they cannot directly capture or optimize diversity in an online learning process. This motivates our proposed prompt-wise and diversity-aware formulation of DAK-UCB.

**2- JRKE/JKD as the Enabling Metrics for Online Diversity Optimization**

We highlighted that JRKE and JKD uniquely admit a two-sample expectation structure that can be rewritten as a single-sample surrogate reward. This property is essential for applying kernelized-UCB confidence bounds and obtaining regret guarantees. Other standard diversity metrics, including Recall, Coverage, and Vendi, lack such a structure and therefore are challenging to optimize efficiently in an online bandit framework.

**3- Prompt-Dependent Mixtures and the Role of Mixture-DAK-UCB**

We highlighted that the diversity-aware optimal policies may require non-degenerate mixtures for each prompt. Unlike fidelity-based methods, where the optimal mixture of the models collapses to a single arm, Mixture-DAK-UCB learns prompt-conditioned mixtures in optimizing diversity that exploit complementary model biases. We discussed that our experiments with Llama/Gemma/Qwen on city and celebrity prompts show considerable gains in diversity of the found mixture of the models compared to the best individual model.

**4- Embedding-Agnostic and Fidelity-Agnostic Framework**


We clarified that DAK-UCB can operate with any choice of text/image embeddings or fidelity score $\phi(t,x)$. While CLIP and DINOv2 are used in our experiments following the recommendations of the references (Hessel et al., EMNLP 2021) and (Stein et al., NeurIPS 2023), the DAK-UCB method can function similarly with other embeddings, such as OpenCLIP, SigLip, and other alternatives. Furthermore, our blur-based numerical analyses show that the chosen embeddings do not spuriously reward noise, validating their application for diversity-aware model selection.

**5- Additional Empirical and Computational Support for DAK-UCB**

In response to the reviewer’s question on the computational costs of DAK-UCB, we included standard random Fourier feature (RFF)-based implementations of DAK-UCB with RBF kernel to show the scalability of the method by applying the well-known RFF framework. We also provided runtime comparisons of mixture vs. non-mixture variants, additional text/image experiments beyond gender diversity, convergence plots, and a new-arm adaptation experiment. These additional numerical results could adequately address the comments on complexity, convergence, and applicability across domains.

**6- LLM-Based Illustration of Prompt-Aware Diversity Learning**

In the responses and our revised manuscript, we have included an illustrative LLM experiment where Llama, Gemma, and Qwen could collapse with different patterns in generating sentences on cities or celebrities. We numerically demonstrated that Mixture-DAK-UCB automatically identifies and combines these complementary modes, achieving significantly higher conditional Vendi scores. This scenario highlights the practical importance of prompt-aware mixture policies and further illustrates that the diversity-aware optimal solution is not a single model.

---

### Meta-Review · Area_Chair_psgs · 2026-01-06

**Summary:**

Common concerns of reviewers include:
1. The diversity objective might conflict with fidelity (Reviewer 4Gv8, rNZi, E2Mq).
2. Computational cost of the model. For example, Reviewer E2Mq points out the lack of run-time analysis. Reviewer a2p3 is curious about how the proposed method compare with mixture-DAK-CUB. Reviewer 4Gv8 is concerned about high computational overhead in general.
3. Reviewer 4Gv8 and rNZi also expressed concern about novelty of the method.

**Reviewer Concerns:**

The authors provided detailed run-time profiling in response to questions regarding run-time. They also provide an experiment show that the trade-off between fidelity and diversity is not as severe because the proposed method made use of the embedding space. I believe the main and common issues are responded and mostly can be addressed by the authors' rebuttal.

**Reviewer Scores:**

The authors provide strong numerical analysis in responses to reviewer E2Mq and a2p3's concern. I believe they might remain positive. The response address 4Gv8's concern well, so reviewer 4Gv8 might potentially raise score. Reviewer rNZi might have remaining concern about the inherit bias from the feature embedding, so rNZi might have some small chance to maintain negative.

---

### Decision · Program_Chairs · 2026-01-26

Accept (Poster)